# Commitment to honesty oaths decreases dishonesty, but commitment to another individual does not affect dishonesty

Janis H. Zickfeld [1 ✉], Karolina Aleksandra Ścigała [2], Alexa Weiss [3], John Michael [4] & Panagiotis Mitkidis [1]

Social commitment influences our behavior in various ways. Recent studies suggest that social commitment to other individuals or groups can increase dishonest behavior while feeling commitment to moral norms might decrease it. Here we show in a pre-registered series of 7 studies investigating the influence of social commitment on dishonest behavior by sampling 7566 participants across three countries (the UK, the US, and Mexico) that commitment to moral norms via honesty oaths might decrease dishonesty (OR = 0.79 [0.72, 0.88]). To the contrary, we found no credible evidence that social commitment to other individuals increases dishonesty (OR = 1.08 [0.97, 1.20]). Finally, we observed that commitment to moral norms was less effective if participants were committed to another individual at the same time (OR = 0.95 [0.86, 1.06]). Our findings point at the potential effectiveness of honesty oaths, while the observed effect sizes were small compared to previous studies.

[1] Department of Management, Aarhus University, Aarhus, Denmark. [2] Department of Psychology, Aarhus University, Aarhus, Denmark. [3] Department of Psychology, Bielefeld University, Bielefeld, Germany. [4] Department of Philosophy, University of Milan, Milan, Italy. ✉email: jhzickfeld@gmail.com

The Dutch Bankers Oath, introduced in 2015, requires all employees in the financial sector in the Netherlands to commit to putting the interests of their clients and society above their own[1]. The main function of this and other common oaths or pledges, such as the hippocratic or MBA oath, is to commit the oath taker to a moral norm, a shared understanding of what is right or wrong in a given context and community[2]. In general, such social commitment to moral norms has been identified as a potential tool to mitigate unethical and dishonest behavior across different contexts[3,4].

While increased social commitment might have beneficial outcomes when it comes to moral norms, recent studies reveal that social commitment to another individual, by for example performing a joint action or belonging to the same group, can increase dishonesty[4–7]. For instance, the four-eye principle of having two people monitor a certain task, originally intended to reduce dishonesty and corruption, can have backlash effects by increasing dishonest behavior because people might feel strongly committed to a partner in crime[8]. Therefore, social commitment to individuals or groups and social commitment towards moral norms may have opposing effects on dishonest behavior, i.e., the former increasing such behavior and the latter decreasing it. When discussing social commitment to other individuals we also refer to studies that have focused on so-called corrupted collaboration or the dishonesty shift in groups[7,9–11], as these studies have been argued to induce a sense of social commitment[4].

Importantly, there is an ongoing debate about the strength of these inhibitory or facilitating effects of social commitment to other individuals or moral norms on dishonest behavior, as recent studies have reported failed replications[12], observed smaller effect sizes[13], or reported evidence of fraudulent data practices[14]. In a recent meta-study[4], overall effects of commitment to other individuals on dishonesty and the effects of commitment to moral norms on honesty were small and showed high heterogeneity and evidence for publication bias. Therefore, it is yet to be shown how robust these effects are in a systematic preregistered study. Such investigation is of particular relevance given that a sense of commitment in organizational settings, to other individuals within the organization or the organization itself, is related to positive outcomes, such as increased job satisfaction or performance[15–17]. Hence, it seems undesirable or even unfeasible to reduce the reliance on work teams to combat dishonest collaboration. Meanwhile, it might be advisable to foster a commitment to specific moral or social norms of honesty to counteract the potential effects of collaborative dishonesty. So far, there is limited evidence on the impact of commitment to moral norms in the context of feeling committed to other individuals or groups[18].

Herein, we provide a systematic investigation of the effects of two different forms of social commitment: commitment to (1) other individuals and (2) moral norms (both alone and in combination) on dishonest behavior across various methods, measures, and populations. The present studies explore the joint effects of a sense of commitment to other individuals and commitment to moral norms on dishonest behavior[18,19]. Our findings shed light on the interplay of these factors on dishonest behavior, providing possible guidelines and tools on how to reduce dishonesty in an organizational setting and beyond.

Social commitment has been defined in various ways, focusing on specific commitments to other individuals[20] or commitments to organizations[15,21]. Here, we adopt a minimal definition of social commitment arguing that social commitment refers to the dispositional state of an agent (X), who is motivated to carry out an action (z) because some other agent(s) (Y) is relying on them to do so[4,22]. Thus, social commitment can refer to a specific action or a general motivational dispositional state of agent X in relationship to agent Y. It is also sufficient that X thinks that Y is relying on the specific action. In contrast to self-commitment (e.g., committing to the goal of eating less candy), social commitment always involves at least two agents. Social commitment can further be described as unilateral, with only one agent feeling committed to the other, and mutual, with both agents feeling committed to each other by the same (joint) or different (complementary) goal[23].

As an example, Panos (X) might feel motivated to finish an important report (z) because his supervisor John (Y) is relying on him to complete the task, creating a social commitment to John on the part of Panos. In the case of only Panos being committed to John, this would classify as a unilateral commitment. However, John (Y) might be motivated to pay (w) Panos (X) to finish the report because Panos is relying on this, creating a mutual complementary commitment. In a different scenario, both Panos and John might need to work on the report, creating a joint commitment.

We argue that although specific actions and contexts might differ substantially, both commitment to other individuals, by for example being part of the same group or working on the same task, and commitment to moral norms, by for example completing an honesty oath, can be considered a basic form of social commitment[4].

For instance, in the case of social commitment to other individuals, a common paradigm to assess dishonesty is the sequential die roll task[9] in which two individuals sequentially roll a die in private and receive a reward if both their reports match. Participants have an incentive to behave dishonestly by coordinating their reports. In this case, Player A (X) is motivated to report a high number (z) and Player B (Y) relies on this action to maximize their outcome. Similarly, Player B (Y) is motivated to match Player A's report (w) and Player A (X) relies on this action to maximize their outcome. Thus, the task should induce complementary social commitment between the two individuals. As the die roll is determined randomly and occurs privately, they are incentivized to misreport their behavior to increase their own and their partner's outcome. Manipulating the interdependence of outcomes such as in the sequential die roll task should increase the motivational state to perform an action (i.e., maximizing outcome) because another agent relies on it, that is social commitment[4] This is likely to highlight prosocial, collaborative, or loyalty norms[24]. Similarly, repeated performance as in the sequential die roll task might further strengthen the social commitment. Performing the action z and Y reciprocating with w, should further increase the motivation to perform action z (and thereby also social commitment to Y)[7].

In the case of social commitment to moral norms, a typical task is asking participants to commit to an honesty oath[3,25,26]. In this case, the participant (X) takes the honesty oath by for example signing, checking, or copying it[27] and is motivated to accurately report their outcome in a task (z) since the party representing the oath (Y), in the context of experimental studies commonly the experimenter, relies on this action. This creates a unilateral commitment to the experimenter on the part of the participant. The experimenter relies on the participants accurately reporting their outcomes because dishonest behavior on the part of the participant will result in financial losses (and violate a social norm of honesty). Manipulating the commitment to a moral norm (e.g., via an honesty oath) should increase the motivational state to perform an action (i.e., reporting accurately) because another agent relies on it, that is social commitment[4]. Of course, it can be argued that committing to an honesty oath also represents a form of self-commitment in which individuals commit to their future self in telling the truth. However, we emphasize that honesty

oaths represent social institutions[28,29] and that nonobservance has direct negative consequences on other individuals, organizations, or society at large[30]. Therefore, commitment to an honesty oath can be considered as a commitment to a social norm, as other individuals in a given community or society expect and demand of each other to follow it and do so with a certain prevalence[31]. This emphasizes the view that social commitment to moral or social norms is embedded in every social, cultural, and interpersonal environment[4]. We will now turn to the question of how social commitment can influence dishonest behavior.

While dishonest behavior, here defined as intentionally misreporting private information[32], is widespread[33,34] and destructive for the society at large[35], people typically do not cheat to the full extent possible[36]. It is argued that individuals strive to maintain a positive self-concept, and that acting dishonestly may threaten such a concept and induce psychological tension[37,38]. To reduce this tension, people might either justify their dishonest behavior or refrain from it.

When feeling commitment to other individuals, there might be a dilemma insofar as one is inclined to act prosocially and loyal towards the other individuals, on the one hand, and an inclination to adhere to general moral principles on the other hand[39]. When the prosocial norm aligns with salient moral norms, cooperating with another individual appears to be the unequivocally right thing to do. Nevertheless, sometimes cooperating or feeling committed to another individual might threaten what is generally considered as injunctively moral (e.g., in the case of corruption[40,41]). It has been argued that acting dishonestly might be more easily justified if it not only benefits oneself, but also another person[42–44]. Based on a recent meta-analysis[7], it was found that prosocial concerns were indeed important in driving dishonest behavior in group settings. Further, it has been argued that being committed to other individuals and interacting can increase the likelihood of communicating and being made aware of possible dishonest strategies[10,45]. Similarly, it has been argued that commitment to other individuals can increase the likelihood of being exposed to dishonest people[46]. Scholars also suggested that commitment to other individuals increases a possible diffusion of individual responsibilities, avoiding potential individual moral concerns or negative emotions[43,47].

Indeed, empirical and meta-analytical evidence suggests that social commitment to other individuals can increase dishonesty[4–7] and might be one of the driving forces behind large-scale corruption[48]. A recent meta-analysis[4] observed an effect of $g = -0.17$ [$-0.25, -0.10$] across 215 studies. At the same time, this effect showed high heterogeneity across studies and evidence for publication bias, suggesting that the effect might be moderated by the type of paradigm, whether the act occurs in public or private, and who is the victim of the dishonest act. Similarly, another meta-analysis[7] identified the location of the study (whether it was conducted in the lab or online), whether participants were deceived, and the type of reward as moderators.

In contrast, committing to the moral honesty norm (e.g., by pledging an oath, as investigated in numerous studies[3]), can reduce justification processes[26,49] by making the responsibility for one's own actions salient[30], reminding the individual of the morally right decision and thereby reducing the likelihood of dishonest behavior. Similarly, it appeals to image concerns of being perceived as an honest and consistent person, someone who behaves according to their word[38,50], while also making honest responses more intuitive[25]. Findings suggest that committing to an honesty oath reduces decision time for reporting the correct choice[25,51].

Indeed, further empirical evidence supports the assertion that social commitment to moral norms decreases dishonesty and increases honesty[18,26,45,52–55]. A meta-analysis[4] found that

honesty oaths or pledges can increase honest behavior ($g = 0.24$ [0.16, 0.32]). Again, they found high heterogeneity and evidence for publication bias. In addition, field studies have provided mixed to no support for the idea that commitment to moral norms can increase honesty[56–58].

Considering all the previously mentioned points, why would different forms of social commitment result in various outcomes with regard to honest behavior? The two types of commitment are likely to make two different norms salient: a moral concern for honesty (i.e., wanting to be an honest person[3]) and a prosocial and loyalty concern targeted specifically at the other individual (i.e, wanting to cooperate with the other individual(s) and wanting to behave loyal to the other individual(s)[24,]). When the concern for cooperation collides with the concern for honesty, individuals might be willing to trade their moral currencies, and prioritize prosocial concerns, in order to justify their dishonest behavior, especially if they can gain from it[39]. We expect that a stronger commitment to other individuals should increase attendance to prosociality norms and concerns, whereas a stronger commitment to a general moral norm (e.g., telling the truth) should increase attendance to moral norms and concerns. We acknowledge the fact that prosocial or loyalty norms are often considered moral or the right thing to do (e.g., it is considered morally right to altruistically help another person). However, we focus on situations in which the prosocial norm (i.e., helping another person or group) goes against the moral norm (i.e., telling the truth).

Similarly, it is likely that social commitment to individuals and moral norms affects felt responsibility for one's actions differently. Whereas feeling committed to moral norms should increase felt responsibility, this should be reduced for commitment to other individuals. Based on this theoretical argumentation and previous findings, we predict that different types of social commitment should have opposing effects on dishonest behavior:

H1. Social commitment to other individuals (compared to the baseline) increases dishonest behavior.

H2. Social commitment to an honesty oath (compared to the baseline) decreases dishonest behavior.

For social commitment to other individuals, we mainly focus on mutual commitment in dyads in the current studies. That is, both individuals in a dyad commit to each other and benefit from possible dishonest behavior. Social commitment has also been investigated in groups and in ways that the outcome is not contingent on the other's behavior[10]. For social commitment to moral norms, we focus on an honesty oath to test commitment to moral norms, as done in previous studies[53,59]. Honesty oaths have been categorized as external commitments that appeal to the internalization of norms[3].

Interestingly, and likely closer to real world conditions, we asked: what happens when people are committed to both other individuals and moral norms? Does one commitment override the other? Two previous studies[18,19] have explored the effect of moral reminders on cheating in team settings compared to individual settings. The first study[18] found that signing an honesty oath reduced subsequent individual dishonesty. In addition, the study showed that this effect is similar if signing an oath is employed for individuals or dyads making decisions. Importantly, collaborating in dyads did not significantly increase dishonesty in contrast to working alone in the first place when no oath was in place. Thus, this study does not give us any strong indication of whether committing to a moral norm would reduce the effect of collaborative corruption (as it was not observed in the first place). The second study[19] found that individuals in groups of three were slightly less likely to behave dishonestly when they also committed to an honesty oath, but this effect was stronger when no payoff commonality (i.e., participants only got a reward if all

three groups members reported the same outcome) was present. This result might suggest that commitment to an honesty oath is less effective in reducing dishonesty when social commitment to other individuals is strong. Importantly, this study did not investigate the effects of commitment to an honesty oath in an individual context, and cheating rates in the group context were significantly stronger compared to the individual baseline treatment. Thus, both studies found mixed evidence of the effect of simultaneous social commitment to another individual and a moral norm compared to a baseline without any commitment.

From a theoretical perspective, we would expect that social commitment to other individuals and moral norms at the same time, induces attendance to both prosocial norms on the one hand (i.e., wanting to help the other individual or the group) and attendance to moral norms on the other hand (i.e., wanting to be an honest person). Whether one of these can override the other might depend on the salience of each concern or norm (i.e., the strength of the (social) commitment). In the discussed meta-analysis[4] both effects were similar in strength ($g = -0.17$ vs. $g = 0.24$), suggesting that combining both commitments might cancel each other out. Therefore, we predict that:

H3. Combining a social commitment to individuals and to an honesty oath (compared to the baseline) does neither increase nor decrease dishonest behavior (null effect).

In sum, we predict that social commitment to an honesty oath reduces dishonest behavior, commitment to an individual increases dishonest behavior, while the combination of commitment to an oath and an individual does neither increase nor decrease dishonest behavior.

## Methods
**The present studies**. We investigated the effects of social commitment on dishonest behavior across seven preregistered online studies across three populations (the UK: Study 1–4; Mexico: Study 5, the US: Study 6–7). We selected the three populations to investigate the generalizability of our effects. The three countries were chosen out of convenience. More detailed information on their differences is provided in Supplementary Table 65. The studies followed similar setups, which is why we present them together and note their differences. As in previous studies, we manipulated social commitment to other individuals by using a minimal group approach (Study 1 & 4–5[60]), a paradigm in which rewards are split equally between partners (Study 2–6[37,42,43]) and paradigms allowing for interactive coordination among participants (Study 6–7[9,10]). To induce commitment to moral norms, we presented participants with the opportunity to sign an oath, highlighting the norm of honesty (Study 1–7). This paradigm has been effectively used across different contexts, such as in the field when measuring tax evasion or in online studies[26,49,51–53,55,59]. Our studies therefore provide close and conceptual preregistered replications of previous research focusing on social commitment and oath effects, while an additional contribution is investigating the combination of commitment to individuals and commitment to moral norms across all studies.

We assessed dishonest behavior by using a variant of the so-called mind game (Study 1–5[49,61,62]) or a die roll task (Study 2–3 & 6–7[63]). In both tasks, participants have the opportunity to misreport their performance in order to maximize their payoffs. In both variants, the incentive for misreporting is high, as the actual choice occurs in private and cannot be verified by the researcher[34]. These paradigms deliberately do not allow for measuring dishonesty on the participant-level due to using a private task (but measure dishonesty on the group-level), as previous research has shown that employing public or traceable paradigms substantially reduces cheating rates[64]. Mirroring the

current literature[9–11], we implemented different—commonly used—versions of these tasks, allowing participants to accumulate their payoffs across four (Study 1–2) or ten trials (Study 3), randomly choosing one payoff from four (Study 4–5) or ten trials (Study 7), or using a one-shot task (Study 4-6). In this way, we were able to systematically investigate the impact of different game versions on the propensity to act dishonestly.

**Open practices statement**. We report how we determined our sample size, all data exclusions (if any), all manipulations, and all measures in the studies. All seven studies were preregistered prior to data collection (https://osf.io/sxbfn/registrations; Study 1: 16/11/21; Study 2: 19/01/22; Study 3: 03/02/22; Study 4: 24/03/22; Study 5: 14/04/22; Study 6: 13/04/22; Study 7: 27/04/23). Pre-registrations, study materials, syntaxes, and data files are publicly available on our project page: https://osf.io/sxbfn/.

Informed consent was obtained from all participants prior to performing the study. All studies were reviewed and approved by the ethical review board at Aarhus University (2021-103; 2022-013; 2022-073). The studies were conducted between 2021 and 2023 (Study 1: 16/11/2021-23/11/2021; Study 2: 19/01/2022-07/02/2022; Study 3: 03/02/2022; Study 4: 24/03/2022- 28/03/2022; Study 5: 14/04/2022 - 21/04/2022; Study 6: 13/04/2022 - 14/04/2022; Study 7: 27/04/2023-29/04/2023).

**Participants**. An overview of the sample size determinations, sampling locations, exclusions, the demographic composition, and the final sample sizes is provided in Tables 1 and 2. Table 1 provides a specific overview of the power analysis and exclusion criteria for each study separately and Table 2 gives an overview of the different designs including the final sample size, and the independent, and dependent variables.

We performed a-priori power analyses based on different effect sizes found across the seven studies. A detailed overview is provided in Table 1, Supplementary Methods, and Supplementary Note 1. For Study 1, we based the sample size on the smallest effect size (the effect of social commitment to individuals on honesty, $g = -0.22$) as reported in the previous meta-analysis[4]. In Study 2, we focused on the meta-analytic effect from this meta-analysis for the specific paradigm used (for investment/effort tasks, $g = -0.19$). Note, that these effects were based on the first version of the meta-analysis. After revising the meta-analysis, they reported a smaller effect size for the effect of social commitment to individuals on honesty, $g = -0.17$. Similarly, the effect size for the investment/effort tasks was $g = -0.15$. Study 3 used the oath effect size obtained in Study 1 and 2. In Study 4 and 5 we used the effect size for the oath treatment from Study 3. In Study 6, we employed the effect size for the oath treatment from Study 4. Finally, for Study 7, we used our smallest effect size of interest ($d = +/-0.15$). Power calculations were performed focusing on an ordinal response using the posamsize function of the hmisc package (version 4.4-1[65]). Relative frequencies of responses were based on previous studies as detailed in Table 1. Based on these results, suggested sample sizes ranged between 1060 and 7408 participants when considering four treatments in total. Due to resource constraints and the fact that except for Study 6 all studies employed repeated measurements and therefore an increased number of observations, we registered a sample size of 800 in Study 1, 1000 in Study 2, and 500 in Study 3. For Studies 4–7, we registered a sequential analysis approach in order to adopt a more efficient way to conduct a high-powered study given the small effects of the previous studies and to save resources[66]. Based on the expected effect size, we set a maximum sample size at 1600 and registered four sequential analysis steps (i.e., analyzing the data at 400, 800, 1200, and 1600 participants)

**Table 1 Overview of sample size determinations, sample sizes, exclusions, and final *n* across the seven different studies.**

| Study | Power Analysis Effect | Effect Source | Suggested Sample Size[g] | Planned Sample Size (observations) | Sample N (obs) | Exclusion Criteria (n)[e] | | | | | | | | | | Final N (obs) |
|---|---|---|---|---|---|---|---|---|---|---|---|---|---|---|---|---|
| | | | | | | Not consenting | Failed Understanding | Failed Attention Check | < 18 | Speeders/ Slow responders[z] | Not correctly committing to oath | Duplicate IDs | Wrong Group | Wrong art preference | Not Matched/ Partner Left | |
| 1 | H1: OR = 0.647[a] | Zickfeld et al. (2023) | 1060 | 800 (6400) | 819 (6576) | 0 | 1 | 0 | 0 | -[h] | 17 | 9 | 17 | 15 | - | 770 (6160) |
| 2 | H1: OR = 0.748[a] | Zickfeld et al. (2023) | 2320 | 1000 (4000) | 2252 (9004) | 116 | 260 | 39 | 5 | 400/99 | 33 | 0 | - | - | - | 1494 (5976) |
| 3 | H2: OR = 0.85[b] | Studies 1-2 | 7408 | 500 (5000) | 509 (5090) | 0 | 4 | 1 | 1 | 5/15 | 1 | 0 | - | - | - | 484 (4840) |
| 4 | H2: OR = 0.74[c] | Study 3 | 2160 | max 1600[f] | 1684 (4230) | 2 | 37 | 2 | 0 | 44/19 | 2 | 42 | - | - | - | 1541 (3896) |
| 5 | H2: OR = 0.74[c] | Study 3 | 2160 | max 1600[f] | 1077 (2712) | 1 | 33 | 0 | 0 | 26/12 | 1 | 0 | - | - | - | 982 (2491) |
| 6 | H2: OR = 0.72[d] | Study 4 | 1840 | max 1600[f] | 835 | 0 | 40 | 0 | 1 | 29/22 | 0 | 0 | - | - | - | 755 |
| 7 | H2: OR = 0.76[a] | SESOI | 2722 | max 2000[f] | 1607 (24710) | 0 | 21 | 0 | 0 | 12/- | 13 | 1 | - | - | 21 | 1540 (23830) |

*OR* odds ratio, *SESOI* smallest effect size of interest.
[a]distribution of scores in the die roll was based on Weisel & Shalvi (2015).
[b]frequencies of scores were based on Study 1 because it was based on the same population (Prolific).
[c]frequencies based on Study 3.
[d]frequencies based on Study 4.
[e]Note, that some participants are excluded by several criteria, thus summing all exclusion criteria does not necessarily lead to the total number of exclusions. Also consider that participants not consenting or failing to understand the task did not complete the main survey so they also quite likely count as speeders.
[f]We applied a sequential analysis (Lakens, 2014), we registered to analyze the data at 400, 800, 1200, and 1600 participants (800, 1200, 1600, 200 in Study 7) continuing if the two main hypotheses (H1 & H2) were not statistically significant. We controlled for the Type-1 error rate by adjusting the alpha level at each step based on a power family function using the GroupSeq package in R. Selecting four interim times, two-sided bounds, suggested an alpha level of 0.0125, 0.0161, 0.0203, and 0.0248 at the four possible different samples.
[g]all sample sizes 80% power, alpha 0.05.
[h]not registered in Study 1.
[z]speeders: 1/3*median duration; slow responses: 3* median duration.

**Table 2 Overview of population, demographic composition, sample size, type of independent/dependent variables, and covariates across the seven different studies.**

| Study | Location | N | | | | | Age | | | Design | Independent Variable | | | Dependent Variable | | | | Covariate |
|---|---|---|---|---|---|---|---|---|---|---|---|---|---|---|---|---|---|---|
| | | Total | w | m | non-binary | n.s. | Range | M | SD | | Com. to Partner | Mani Check | Com. to Norm | Task | n rounds | Max Bonus Payoff (Base Payment) | $M_{bonus}$ (SD) | HH (α) |
| 1 | Prolific, UK | 770 | 379 | 386 | 3 | 2 | 18-80 | 41.19 | 13.44 | 2 (commitment to individual (ci):between(b)) x 2 (commitment to moral norm (cm):b) x 2 (target: within (w)) mixed | ART Task | IOS | written Oath[b] | Mind Shape | 8 (4 for self, 4 for other) | £4 (£2) | £2.43 (£.55) | 0.74 |
| 2 | Toluna, UK[a] | 1494 | 893 | 594 | 6 | 1 | 18-91 | 53.66 | 15.72 | 2 (cib) x 2(cm:b) x x 2 (task:b) between | Shared Payoff[x] | IOS, Com. (pre-post) | | Mind Shape/ Die Roll | 4 | 1200pts[c] (1200pts) | 652.10pts (204.82) | 0.73 |
| 3 | Prolific, UK | 484 | 242 | 240 | 2 | - | 18-78 | 39.81 | 13.25 | | | | | Die Roll | 10 | £2 (£1) | £1.19 (£.27) | 0.74 |
| 4 | Prolific, UK | 1541 | 750 | 775 | 9 | 7 | 18-93 | 41.24 | 13.84 | 2 (cib) x 2(cm:b) x 2 (length of task:b) between | Chat + Group Name, Shared Payoff | | | Mind Shape | 4/1 | | £1.35 (£.56) | 0.74 |
| 5 | Prolific, Mexico | 982 | 480 | 478 | 22 | 2 | 18-63 | 25.66 | 6.49 | | | | | | | | £1.32 (£.53) | 0.69 |
| 6 | Prolific, US | 755 | 369 | 370 | 11 | 5 | 18-78 | 39.46 | 12.82 | 2 (cib) x 2(cm:b) x 2 (type payoff:b) nested | Chat, Shared Payoff/Double | | | Die Roll | 1 | | £1.35 (£.66) | 0.80 |
| 7 | Prolific, US | 1540 | 748 | 763 | 23 | 6 | 18-78 | 41.12 | 13.27 | 2 (cib) x 2(cm:b) x | Sequential Die Roll (Shared Payoff/ Double) | | | | 10 | £1.80 (£1) | £0.42 (£0.67) | 0.59 |

Note that the amount of points was set by the survey company according to the estimated length of the survey.

w women, m men, n.s. not-specified, com. to partner commitment to partner, com. to norm commitment to social norm, mani check manipulation check, ART task art preference task (Tajfel et al., 1971), IOS inclusion of the other in the self scale (Aaron et al., 1992), HH honesty-humility.

[a]The sample was originally intended to be representative of the population by gender, age, and region. However, due to a technical error, some combinations were oversampled, which resulted in a sampling of additional participants to fill up the remaining combinations.

[b]The oath read: I promise that the information I am providing in this study is true (in Study 1–6) and: I hereby declare to provide honest information in this study (in Study 7).

[c]According to Toluna 5500 pts = £1 (1200 pts = 0.22 p).

for Study 4-6, and set a maximum sample size of 2000 and four sequential steps (800, 1200, 1600, 2000 participants) for Study 7. As preregistered, recruitment was stopped once the main effects were statistically significant. In order to control for the Type-I-error rate[66], we adjusted the alpha level at each step based on calculations using the GroupSeq package (version 1.4.0[67]; see Table 1 for more detailed information). In Study 4, we collected the full number of 1600 participants. Only in two cases, Study 5 and 6, we stopped data collection preliminarily, deviating from our registered plans in the following ways: In Study 5, we stopped during the third wave because we were not able to collect enough participants at the same time due to a restricted participant pool, which was essential for the interactive nature of the task potentially leading to a high number of drop-outs. In Study 6, we stopped after the second wave (i.e., 800 participants) although the main effects were not statistically significant. However, we realized that the effects were either tiny or in the opposite direction compared to the previous studies. We think that these changes were sensible and do not alter the overall patterns and findings in any substantial ways. In Study 7, we stopped data collection after 1600 participants according to our preregistered analysis plan, as our main effects were statistically significant.

As final sample sizes across all studies were smaller than the suggested sample sizes by our power analyses, we conducted post-hoc sensitivity analyses for all studies in order to investigate what effect size we could minimally detect at a power of 90% and 95% (see Supplementary Note 2 and Supplementary Table 1). Across studies, the minimum effect size we could detect at 90% power ranged between $d = -0.09$ and $-0.19$ and between $d = -0.10$ and $-0.22$ for 95% power. The minimum effect size was larger in Study 6 (90% $d = -0.30$; 95% $d = -0.34$) due to the fact that we employed a one-shot game. These effect sizes are typically considered as small in the literature[68] and are in the range of our smallest effect size of interest ($d = +/-0.15$). In addition, the meta-analytic investigation across 7576 participants helps us to increase our power to detect even smaller effect sizes, such that we can be certain that we have enough power to detect our smallest effect size of interest when combining all studies.

We recruited participants via the crowdworking website Prolific.co[69] in Study 1 and Studies 3-7 and via the panel provider Toluna in Study 2. Studies 1-4 sampled participants in the UK, while Study 5 sampled participants in Mexico, and Study 6-7 participants in the United States. While we focused on place of residence when recruiting participants, the clear majority of participants (i.e., 90-98%) reported the respective citizenship associated with the specific geopolitical region (see Supplementary Methods). All participants received a base study compensation according to the length of the study and were able to receive an additional bonus payment on top based on their performance in the game task (see Table 2 for an overview of compensations).

Across studies, our exclusion criteria were: not consenting to start the study (these were excluded automatically), not understanding the main task based on a probe item, failing an attention check item, being younger than 18, responding significantly faster or slower compared to the median response time as registered, not correctly committing to the oath (by writing their name or pasting their participant ID; note that not committing to the oath did not result in an exclusion), being duplicates of prior participants based on the ID assigned by the crowdsourcing service, or not being matched in the interaction task or the matched partner failing to respond (see Table 1 for detailed definitions). In Study 1, participants were also excluded because they did not state the correct art preference based on the minimal group manipulation (e.g., by naming a different artist). Note, that not correctly committing to the oath and being a duplicate ID were not preregistered as exclusion criteria originally. However,

we noticed such behavior in Study 1 and argue that it represents a sensible addition. Sensitivity analyses without such exclusions are presented in Supplementary Table 5-9 & Supplementary Fig. 6.

After applying all exclusion criteria, we recruited a total of 7566 participants across seven studies (3859 women, 3608 men, 76 non-binary, 23 not specified) ranging from 18 to 93 years ($M = 41.3$, $SD = 15.4$; see Table 2 for sample sizes for each study separately).

**Procedure**. All studies featured a fully crossed 2 (commitment to individual: individual vs. partner) x 2 (commitment to moral norm: no oath vs. oath). Throughout the text we will refer to the individual cells as baseline (individual - no oath), partner (partner - no oath), oath (individual - oath), and partner+oath (partner - oath). Specific operationalizations differed across studies (see Table 2). In addition, the different studies varied on the type of the economic game task and the number of rounds (1, 4, 8, or 10 trials), the possible bonus payoff, and the way the bonus payoff was calculated, in order to provide robustness tests of the main hypotheses and investigate possible moderators (see Table 2; Supplementary Table 10-27 for an overview). Bonus payoffs (based on the tasks) were calculated based on cumulative responses across rounds in Studies 1-3[11], one randomly selected trial for certain participants in Studies 4-5 and for all participants in Study 7[9], and on a one-shot task (e.g., Kocher et al.[10]) in Studies 4-6 (see Table 2 for an overview of bonus payments).

General procedures were similar across the studies. An overview is provided in Fig. 1. Participants always received information about the study and their rights and provided informed consent. Afterward, they were randomly allocated to one of the four social commitment treatments (see Table 3 for cell sizes). In Study 1, participants in the oath and partner+oath treatments were then presented with the oath manipulation and then participants in all treatments completed the Artistic Preference Task (ART[70]). In all studies, participants then received instructions about the specific economic game task including an item probing for understanding of the task. In case of choosing the wrong answer, participants were shown the instructions once more and had another opportunity to respond to the probe item. In Studies 4-5 and 7 in the partner and partner+oath treatments, participants were afterward matched with another partner, and in Studies 4-5 engaged in a chat for up to 3 min before the main task with the objective of deciding on a group name for the dyad – a task that has been successfully implemented in previous studies to induce social commitment[60]. In all except Study 1, participants then completed the inclusion-of-the-other-in-the-self item[71] and a commitment item to assess their commitment towards the matched partner. In Study 2, these measures were only included in the partner and partner + oath treatments due to a technical error. In Studies 2-7, participants in the oath and partner + oath treatments were at this stage presented with the oath manipulation. In all studies, participants then completed the main economic game task. The type of task, number of rounds, and how the final bonus payment was calculated differed across studies as summarized earlier and in Table 2. In Study 6, participants completed the die roll paradigm[63] by first rolling a die in private and then completing a chat for up to 2 min with their matched partner in the partner and partner + oath conditions to discuss their final report, as the individual report influenced the dyad payoff. In Study 7, participants completed a sequential die roll game for ten rounds based on a previous study[9]. In all studies participants then completed two items to assess commitment to the partner, an item on felt responsibility (except for Study 7) and a short questionnaire on trait honesty-humility. In Study 1, participants also completed a questionnaire

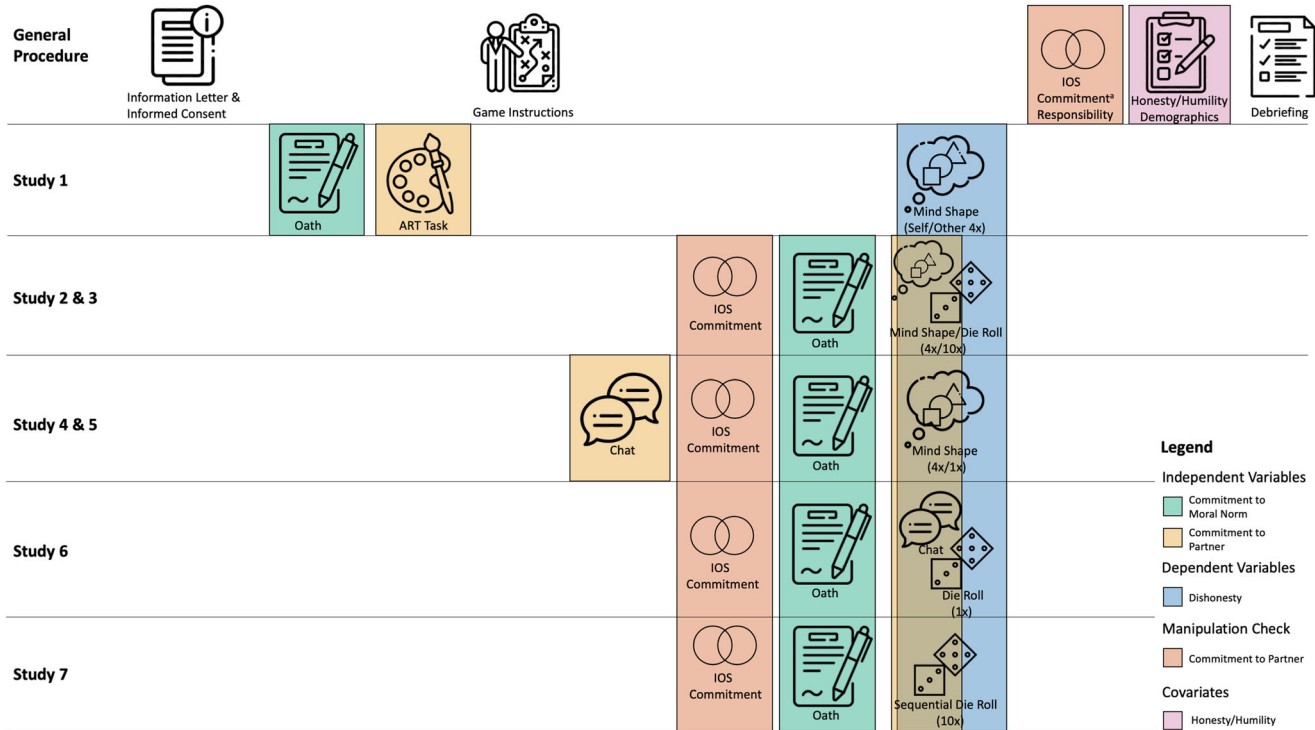

**Fig. 1 Overview of the different procedures and methods across the seven studies.** Schematic overview of the procedures across the seven studies. Each row corresponds to a specific study, while the columns represent the specific steps in the experimental procedure. Colored boxes refer to specific variables in the procedure. Manipulations of the independent variables, commitment to moral norms, and commitment to a partner are represented in green and orange respectively. The dependent variable, dishonesty measured in the economic games is depicted in blue. The manipulation check of assessing commitment to the partner is depicted in red, while honesty-humility as the covariate is depicted in purple. Note that in Studies 2–7 the economic game manipulated commitment to the partner, while also assessing dishonesty, which is why it is depicted in orange and blue (icons: Flaticon.com). ªThe commitment item was included in all studies except for Study 1.

on trait self-control, but this measure was dropped in the later studies as it did not show a statistically significant relationship with dishonesty. Study 1 also included a similarity measure in which participants were asked to arrange the geometrical shapes employed in the mind shape task based on their perceived similarity in a 2D-space[72]. Finally, participants completed demographic information and were then thanked and fully debriefed.

**Commitment manipulations**. In Study 1, we manipulated commitment to the partner by using the artistic preference (ART) task[70,73], as done in previous studies[60,74,75]. Participants were presented with five pairs of paintings, each pair including a painting by Paul Klee and Wassily Kandinsky, and were asked to select which one they preferred. To manipulate commitment to the partner in the partner and partner + oath treatments participants were informed that their matched partner shared the same preference as they did prior to the economic game task resulting in payments for the other. Participants did not interact and actual matching was only performed after the study (see Supplementary Material 1.1 for more details).

In Studies 2 to 6, we manipulated commitment to the partner by using a shared incentive scheme in the partner and partner +oath treatments[42,43]. In the baseline and oath treatments, participants received a bonus payment ($\pi_i$) based on their reported outcome ($p_i$) in the specific task, with final bonus payoffs equaling final outcomes ($\pi_i = p_i$[42]). In the partner and partner + oath treatments, participants were randomly matched with another participant from the same treatments (partner or partner+oath), and the individual bonus payoffs were defined

as the sum of both outcome sums divided by two ($\pi_i = \frac{1}{2}*(p_i + p_j)$). Bonus payoffs were thus identical for both participants ($\pi_i = \pi_j$; following a previous procedure[42]). For example, if Participant A reported an outcome associated with £2 and Participant B reported an outcome associated with £0.8, both participants received a final bonus of $\frac{1}{2} * (2 + 0.8) = £1.4$.

In Studies 6 and 7, we added another incentive scheme that was based on the dyadic die roll paradigm[9]. Here, bonus payoffs depended on whether both participants reported the same outcome ($p_i = p_j$). In this case, the associated outcome was equal to the final bonus payoff ($p_i = \pi_i$) and this was the same for both participants. If participants reported different outcomes ($p_i \neq p_j$), they did not receive any bonus payoff ($\pi_i = 0$). As both participants were able to chat before the final report in Study 6, there was an incentive to coordinate and behave dishonestly. The main difference from the sequential die roll paradigm in previous studies[9] in Study 6 was that matched participants decided on their response at the same time and there was no first or second mover, as this was confined to one round.

We extended this design, by adopting the original dyadic die roll paradigm[9] in Study 7. Participants in the partner and partner+oath treatments were matched with a partner and randomly received the role of the first or second mover. Across ten rounds, the first mover rolled a die and reported their outcome and this information was sent to the second mover who then rolled a die and reported their outcome. Rules were the same as in Study 6, both participants received points if both outcomes matched ($p_i = p_j$) and no points if the outcomes were different ($p_i \neq p_j$). At the end, one round was randomly selected for payment. In the individual treatments (baseline, oath)

**Table 3 Overview of economic game task scores across the six different studies and treatments.**

| Study | Target/Task | Baseline | | Partner/Minimal Group | | | Oath | | | Partner+Oath | | |
|---|---|---|---|---|---|---|---|---|---|---|---|---|
| | | n (obs) | M (SD) | n (obs) | M (SD) | OR [95% CI] | n (obs) | M (SD) | OR [95% CI] | n (obs) | M (SD) | OR [95% CI] |
| 1 | Self | 198 (792) | 4.17 (1.63) | 194 (776) | 4.26 (1.60) | 1.06 [0.84, 1.34] | 188 (752) | 4.09 (1.68) | 0.80 [0.63, 1.02] | 190 (760) | 4.20 (1.63) | 1.10 [0.87, 1.39] |
| 1 | Other | | 3.89 (1.72) | | 3.96 (1.66) | 0.98 [0.86, 1.12] | | 3.71 (1.75) | 0.84 [0.73, 0.96] | | 3.98 (1.70) | 0.93 [0.81, 1.07] |
| 2 | Mind Shape | 892 | 3.72 (1.63) | 788 | 3.74 (1.66) | 0.95 [0.75, 1.20] | 776 | 3.56 (1.64) | 0.74 [0.58, 0.94] | 696 | 3.67 (1.64) | 0.74 [0.58, 0.94] |
| 2 | Die Roll | 768 | 3.85 (1.58) | 704 | 3.79 (1.56) | | 696 | 3.70 (1.66) | | 656 | 3.78 (1.59) | |
| 3 | Mind Shape | 58 (580) | 4.26 (1.59) | 80 (800) | 4.12 (1.66) | 1.10 [0.87, 1.40] | 52 (520) | 3.94 (1.76) | 0.72 [0.58, 0.90] | 62 (620) | 3.95 (1.67) | 0.77 [0.61, 0.97] |
| 3 | Die Roll | 65 (650) | 3.95 (1.67) | 55 (550) | 3.98 (1.64) | | 65 (650) | 3.76 (1.72) | | 47 (470) | 3.76 (1.62) | |
| 4 | Four Trials | 221 (884) | 4.38 (1.58) | 159 (636) | 4.48 (1.54) | 1.31 [1.05, 1.64] | 226 (904) | 4.15 (1.64) | 0.76 [0.61, 0.94] | 179 (716) | 4.16 (1.649) | 1.11 [0.89, 1.38] |
| 4 | One-Shot | 211 | 4.59 (1.52) | 207 | 4.59 (1.58) | | 180 | 4.41 (1.47) | | 158 | 4.48 (1.55) | |
| 5 | Four Trials | 131 (524) | 4.12 (1.64) | 116 (464) | 4.12 (1.64) | 0.96 [0.67, 1.37] | 131 (524) | 3.90 (1.62) | 1.16 [0.82, 1.65] | 125 (500) | 4.23 (1.55) | 0.96 [0.68, 1.37] |
| 5 | One-Shot | 123 | 4.38 (1.50) | 115 | 4.38 (1.50) | | 126 | 4.14 (1.49) | | 115 | 4.43 (1.56) | |
| 6 | Shared Payoff | 237 | 4.64 (1.43) | 84 | 4.57 (1.43) | 1.15 [0.96, 1.37] | 191 | 4.69 (1.51) | 0.82 [0.72, 0.93] | 75 | 4.37 (1.37) | 0.99 [0.83, 1.19] |
| 6 | Double | | 0.32 (0.47) | 80 | 4.61 (1.51) | | | | | 88 | 4.84 (1.38) | |
| 7 | Report[a] | 419 | 3.83 (1.75) | 356 | 3.97 (1.74) | 1.35 [1.00, 1.83] | 424 | 3.68 (1.75) | 0.72 [0.57, 0.91] | 344 | 3.86 (1.71) | 1.24 [0.92, 1.66] |
| 7 | Double[b] | 419 | 0.32 (0.47) | 334 | 0.37 (0.48) | | 424 | 0.27 (0.44) | | 360 | 0.36 (0.48) | |

[a] refers to die roll reports; for the partner and partner + oath we only focus on responses by Player A.
[b] refers to double reports; for the partner and partner + oath we only focus on responses by Player B (they are identical to Player A, each dyad can report one double response per round).
OR odds ratio, obs observations. Economic game task scores range between 1 and 6. Effect sizes (odds ratios) are reported across the specific target/task manipulations. Effect sizes for Study 1 compare the other targets only.

payoff schemes were the same, but participants took the roles of both first and second mover and thus rolled a dice twice in each round and received one payment in the end, as in the original study[9].

In Studies 1–3, participants did not interact with their partners but were only matched after the study. In Studies 4 and 5, participants interacted with their partner before the main task via a live chat, and in Study 6 participants interacted with their partner via a live chat during the main task. In Study 7, participants engaged in a sequential game task, but they did not interact through a chat and could not communicate as in the original paradigm[9]. Thus, the different studies featured various forms of interaction between participants.

In Studies 4–7, we implemented a live matching and interaction in the partner and partner + oath treatments using the SMARTRIQS framework[76]. Participants in these treatments were randomly matched with another participant from one of the same treatments, which determined their partner for the rest of the study. Note, that participants were not able to tell whether they were in the partner or partner + oath condition in Studies 4–5 as the only difference between conditions (the oath) was implemented after the partner chat. In Study 6, inspecting chat logs verified that no participant communicated this. In Study 7, participants did not engage in a chat. In Study 7, 82 groups consisted of dyads that were both in the partner treatment, 85 groups of dyads in the partner+oath treatment, and 190 mixed dyads (see more detailed analyses Supplement Table 66–67 & Supplementary Fig. 22).

After being successfully matched, participants received instructions that they would engage in a chat for up to 3 min (Studies 4–5) or 2 min (Study 6) with their partner and that they could exit the chat before the time was up. Due to a technical error, chat duration was not recorded. Based on chat logs it is highly likely that the majority of dyads exited the chat before the three or two minutes were up. In Studies 4 and 5, in order to strengthen commitment to the partner, participants were given the task to come up with a group name during the chat[60], and instructions referred to the chosen name throughout the study (see Supplementary Methods 1.7 for more details).

In order to manipulate commitment to moral norms in the oath and partner+oath treatments, we implemented an oath that was identical across studies (with a slight rewording in Study 7). The only difference was that in Study 1 the oath was presented immediately after the information letter, while in Studies 2–7 it was presented right before engaging in the economic game task. Following previous studies[51,77], participants were asked to commit to the statement: Participants in this study commit to the norm of telling the truth. I promise that the information I am providing in this study is true. The first sentence was added in order to make a social norm of honesty salient among the participants and to provide a reason why participants would be asked to commit to such an oath. In Study 7 we dropped the first sentence and slightly reworded the main oath to: I hereby declare to provide honest information in this study. As signing in online studies is difficult (also considering privacy concerns), participants were asked to write the second part of the statement into a text field[77], which has been found to be comparable to signatures in terms of effectiveness[27]. It was not possible for participants to just copy the statement and they had to actively type it in to strengthen commitment. In addition, participants could advance without typing it as voluntary actions have been found to strengthen commitment[51] (importantly the majority of participants committed to the oath as we observed compliance rates between 94.34% and 100% of retained participants, see Supplementary Methods).

**Dishonesty measures**. In order to measure dishonesty behavior, we adapted a modified mind game task in Studies 1–5[62]. In such mind game paradigms, participants are typically asked to think of a certain outcome, then presented with an actual outcome, and asked whether their imagined outcome matches the actual outcome. As individuals are incentivized if both outcomes match, there exists a propensity to cheat. In addition, cheating is non-detectable as it occurs in private, which is one reason that mind game paradigms might induce higher cheating rates in comparison to other commonly used tasks[34]. Here, we introduce the mind-shape task, a mind game task with an ordinal (instead of binary) payoff structure with six different options. In Study 1, the payoff per trial ranged from 0 to 50 pence (0p, 10p, 20p, 30p, 40p, 50p). In Study 2, the payoff structure ranged from 0 to 300 points (0pts, 60pts, 120pts, 180pts, 240pts, 300pts). In Study 3, the payoff structure ranged from 0 to 20p (0p, 4p, 8p, 12p, 16p, 20p). In Studies 4–6 the payoff per trial ranged from £0 to £2 (£0, £0.4, £0.8, £1.2, £1.6, £2).

In each trial, participants were initially presented with six different geometrical shapes. These were randomly taken from a pool of 11 shapes (Supplementary Fig. 1). Participants were asked to pick one shape privately by thinking about it or writing it down in private. Subsequently, each of the six shapes was randomly associated with a payoff (and participants were aware of this fact). Participants then had to indicate which shape they chose. This task allows for cheating as individuals can choose a shape with a higher payoff than originally selected in order to maximize their payoff. Like other mind games[62], the task does not allow for measuring actual cheating at the individual level, but only on the average level, based on the distribution of responses: We can explore whether responses exceed the expected mean value of 3.5 (assuming a scale from 1 to 6).

There are several advantages to this task. First, the mind game is completely private, and actual selections cannot be recorded (which has been found to increase dishonesty[34]). The original mind game[62] only includes a binary outcome (participants either win or not), which reduces the overall power of the measure and also makes dishonesty an all or nothing decision (you can either cheat or not). We therefore combined the mind game with a payoff scheme similar to the die roll task[63]. Importantly, the mind-shape paradigm does not need external applications to mimic a die roll and relies on basic geometric shapes that are recognized across cultures[78–80] (see Supplementary Fig. 1).

To test the validity of the mind-shape task and to use a more commonly used and validated[34] procedure to assess dishonesty, participants in Studies 2-3 and 6-7 completed a die roll task based on previous studies[9,42,63]. In each trial, participants were asked to privately roll a die by either using an actual die or using an external website. We emphasized that responses from the external website could not be linked to our survey, so die rolls were occurring in private. Participants were then asked to report number of pips they rolled with each being associated with a different payoff (Study 2: 1 = 0pts, 2 = 60pts, 3 = 120pts, 4 = 180pts, 5 = 240pts, 6 = 300pts; Study 3: 1 = 0p, 2 = 4p, 3 = 8p, 4 = 12p, 5 = 16p, 6 = 20p; Study 4–6: 1 = £0, 2 = £0.4, 3 = £0.8, 4 = £1.2, 5 = £1.6, 6 = £2; Study 7: 1 = £0.3, 2 = £0.6, 3 = £0.9, 4 = £1.2, 5 = £1.5, 6 = £1.8), leaving the possibility for participants to increase their payoff by misreporting the die roll.

Ratings for both economic games were always on a scale from 1 to 6, with 1 representing the lowest and 6 the highest possible bonus payout.

**Felt commitment to partner**. We assessed felt commitment using the inclusion-of-the-other-in-the-self measure[71], a scale of seven Venn diagrams increasing in overlap, and an item on commitment to the partner (i.e., How committed do you feel to your partner/another participant in the study) on a scale from 1 (not at all committed) to 7 (very much committed) pre (except for Study 1) and post the main dishonesty measure (Fig. 1). Participants in the partner and partner + oath treatments completed the items with regard to their matched partner and with reference to another participant from the study in the baseline and oath treatments. In Study 1, the item was always with reference to the matched participant (who was an out-group participant in the baseline and oath treatments) and participants only completed the inclusion-of-the-other-in-the-self measure.

**Honesty-humility**. We employed the 10-item Honesty-Humility scale (see Table 2 for individual reliabilities[81]) on a 5-point scale from "strongly disagree" to "strongly agree". We also added one attention check item (i.e., This is an attention check. Please select option "3 - Neutral (Neither Agree Nor Disagree)") among the honesty-humility items. Participants failing this attention check were screened out (see Table 1). In Study 7, participants completed the 4-item Honesty-Humility scale[82].

**Additional measures**. In Study 1, participants also completed the 12-item self-control scale[83] ($\alpha = 0.87$) on a 5-point scale from "not at all like me" to "very much like me" and a measure to rate the similarity of the shapes employed in the mind shape task (Q-SpAM[72]). Across all studies except for Study 7, participants also indicated their felt responsibility for their actions (i.e., How much responsibility did you feel for your actions during the [xx] task?) on a 5-point scale from 1 (none at all) to 5 (a great deal).

We also obtained participant's gender, age, nationality, three items on socioeconomic status, childhood socioeconomic status[84], and partial postcode (not in Study 7) across all studies. Gender was obtained based on information provided by participants and included the answer options female, male, third-gender/non-binary, and prefer not to say. Based on the journal's style guidelines, we refer to women and men throughout the text. No data on ethnicity or other socially relevant groupings were recorded. Detailed materials and questionnaires are provided on the project page (https://osf.io/k73sd/).

For the main registered models, the data met the assumptions of the tests (Supplementary Note 6, Supplementary Table 2, and Supplementary Fig. 2). Note, that these tests were not preregistered.

**Reporting summary**. Further information on research design is available in the Nature Portfolio Reporting Summary linked to this article.

## Results

**Confirmatory analyses**. All models represented in the following confirmatory part were preregistered before conducting the study and deviations are clearly noted as such.

We measured how much commitment participants felt to their matched partners (or a hypothetical other participant from the same experiment in the baseline and oath treatments). Note that due to a technical error, we did not measure commitment in the baseline and oath treatments in Study 2. We observed that prior to the game, participants in the partner and partner+oath treatments reported higher commitment to their matched partners than participants in the baseline and oath treatments toward another study participant across all studies (Table 4). This effect was statistically significant in all studies in which we assessed it except for Study 1 (see Table 4). In Studies 2–7, we also assessed whether commitment increased in the partner and partner + oath treatments after engaging in the economic game

**Table 4 Overview of felt commitment ratings across Studies 1–7 and model tests.**

| Study | Overall Test ($dfs$, $F$, $p$, $\eta^2$) | | | | | | | | Comparisons (with Baseline), $t$ ($df$), $d$ [95% CI], $p$ (adj) | | | Comparison Pre-Post, $t$ ($df$), $d$ [95% CI], $p$ (adj) |
| --- | --- | --- | --- | --- | --- | --- | --- | --- | --- | --- | --- | --- |
| | Treatment | | | | Measurement Time | | | | Partner | Oath | Partner +Oath | Pre-Post |
| 1 | 3766 | 2.16 | 0.092 | 0.01 | - | - | - | - | 1.93 (389), 0.20 [−0.001, 0.40], 0.273 | 0.56 (383), 0.06 [−0.13, 0.25], <0.999 | 2.12 (379), 0.22 [0.02, 0.42], 0.206 | - |
| 2 | 1708 | 3.78 | 0.052 | 0.005 | 1708 | 31.29 | <0.001 | 0.002 | - | - | - | 5.59 (710), 0.21 [0.13, 0.29], <0.001 |
| 3 | 3480 | 20.92 | <0.001 | 0.12 | 1241 | 10.02 | 0.002 | 0.002 | 6.45 (237), 0.81 [0.53, 1.10], <0.001 | 1.54 (233), 0.20 [−0.05, 0.47], 0.250 | 6.52 (230), 0.86 [0.59, 1.18], <0.001 | 3.15 (243), 0.20 [0.07, 0.34], <0.001 |
| 4 | 31,537 | 107.9 | <0.001 | 0.17 | 1700 | 11.78 | <0.001 | 0.002 | 13.5 (793), 0.95 [0.81, 1.12], <0.001 | 1.59 (829), 0.11 [−0.02, 0.24], 0.302 | 14.4 (766), 1.03 [0.87, 1.19], <0.001 | 3.43 (702), 0.13 [0.06, 0.21], <0.001 |
| 5 | 3978 | 91.85 | <0.001 | 0.22 | 1468 | 1.63 | 0.203 | 0.0002 | 11.2 (436), 1.01 [0.84, 1.21], <0.001 | −1.41 (509), −0.13 [−0.29, 0.05], 0.348 | 10.7 (439), 0.96 [0.78, 1.15], <0.001 | 1.27 (470), 0.06 [−0.03, 0.15], 0.204 |
| 6 | 3751 | 58.09 | <0.001 | 0.19 | 1324 | 18.82 | <0.001 | 0.01 | 9.54 (376), 0.96 [0.73, 1.18], <0.001 | −0.31 (406), −0.03 [−0.22, 0.16], 0.988 | 9.78 (387), 0.98 [0.75, 1.23], <0.001 | 4.35 (326), 0.24 [0.12, 0.37], <0.001 |
| 7 | 31,536 | 117.9 | <0.001 | 0.19 | 1695 | 22.62 | <0.001 | 0.004 | 13.4 (748), 0.97 [0.80, 1.13], <0.001 | 0.78 (840), 0.05 [−0.08, 0.19], 0.859 | 13.8 (754), 0.99 [0.85, 1.17], <0.001 | −4.76 (696), −0.18 [−0.25, −0.11], <0.001 |

Adjustment of $p$-values via Holm.

task. We observed a statistically significant increase in ratings in all except Study 5 and 7, with participants reporting higher commitment after compared to before the game task (see Table 4). An overview of commitment scores to the partner for each treatment and study is provided in Fig. 2. In Study 7, participants reported significantly lower commitment after as compared to before the game task (Table 4). We explored this finding in a non-registered analysis and observed a statistically significant moderation by the number of doubles reported across the task. Reporting more doubles in the sequential die roll was positively associated with increased commitment to the partner after the task, especially for Player A (see Supplementary Figs. 3–4).

A distribution of the economic game scores for each study and treatment is provided in Fig. 3 and the mean, standard deviation, cell sizes, and effect sizes for the different treatments are provided in Table 3. Across all studies and treatments, participants on average reported significantly higher ratings compared to what would be expected by chance (except for the oath treatment in Study 2 for the mind shape task and Study 7 for reported doubles; see Supplementary Table 68). Overall, dishonesty differed across studies and treatments.

For each study, we performed a multilevel ordered logistic model using the ordinal package[85]. In this model, we used the task score as the dependent variable, treatment as the predictor, and participants as random effects. The model was fitted with the adaptive Gauss-Hermite quadrature approximation using 10 quadrature points. Due to the fact that participants only provided one response in Study 6, we analyzed the data using an ordinal logistic regression. This was originally misspecified in the respective preregistration. In Study 1, we originally registered slightly different analyses due to a different design (see Supplementary Note 8 and Supplementary Table 3–4). In Study 7, we performed a logistic multilevel model given that the outcome variable (double) was binary.

Importantly, there are basically two different measures of dishonesty in the sequential die roll task in Study 7. First, Player A can misreport and potentially increase incentives by reporting higher die rolls. Second, Player B can misreport by matching Player A's report. Therefore, we have two main outcomes one being the task report and the other whether Player B (or the dyad) reports a double. Here, we focus on double reports (focusing only on Player B or the dyad in the partner and partner + oath treatments) as this is typically considered the main outcome[9]. We report more details on this and additional results focusing on the die roll reports in Supplementary Table 8–9. Results were comparable though somewhat weaker for die roll reports (see Table 3).

We then performed three random effect models using the metafor package[86] based on the coefficients obtained from the individual regression models for each study comparing each of the three main treatments against the baseline. A forest plot including the individual effects per experimental treatment and study is presented in Fig. 4. Note, that the meta-analytical approach was not registered in each individual study.

We found no statistically significant evidence for H1. Although participants in the partner conditions were numerically more likely to engage in dishonesty than in the baseline conditions across studies, this effect was not statistically significant ($logOR = 0.08$, 95% CI [−0.03, 0.18], Cohen's $d = 0.04$, $p = 0.150$). There was no statistically significant heterogeneity across studies, $Q(6) = 8.70$, $p = 0.191$, $I^2 = 32.16$ [0, 86.80], with Study 5 and 7 showing the strongest effect in the expected direction and Studies 2, 3 and 6 effects in the opposite direction (indicating less dishonesty in the partner condition compared to the baseline). In the partner and partner + oath conditions, we observed positive

correlations between rated commitment (to the partner) and reported outcomes in the game task ($r = 0.07$ [0.01, 0.14] for partner and $r = 0.08$ [0.01, 0.14] for partner + oath, see Supplementary Figs. 16–19).

As expected (H2), we observed a statistically significant decrease in dishonesty from baseline for participants committing to the oath, $logOR = −0.23$, 95% CI [−0.31, −0.15], $d = −0.13$, $p < 0.001$. There was no statistically significant heterogeneity across the studies, $Q(6) = 7.08$, $p = 0.313$, $I^2 = 0$ [0, 90.08]. Notably, Study 6 showed an effect in the opposite direction. Finally, we found no statistically significant effect for H3. Although we observed a small decrease in dishonesty from baseline for the combined treatment including both commitment to a partner and commitment to the oath, this effect was not statistically significant compared to the baseline, $logOR = −0.04$, 95% CI [−0.18, 0.09], $d = −0.02$, $p = 0.537$. We observed a statistically significant heterogeneity across studies, $Q(6) = 13.83$, $p = .032$, $I^2 = 59.57$ [0, 92.42]. Studies 3 and 4 found that combining both forms significantly reduced dishonesty, while the effect was not statistically significant in the remaining studies. We performed exploratory non-registered analyses comparing the partner+oath effects directly with the other two treatments and found that participants in the partner + oath treatment reported statistically significantly lower scores compared to the partner treatment, $logOR = −0.13$ [−0.23, −0.02], $d = −0.07$, but significantly higher scores compared to the oath treatment, $logOR = 0.20$ [0.03, 0.37], $d = 0.11$ (see Supplementary Figs. 20–21). As registered, we compared the partner + oath effect against our smallest effect size of interest (SESOI) of $d = +/−0.15$ to test for a null effect (see Supplementary Note 7 for more details on the specific SESOI). Performing equivalence testing using the TOSTER package, we observed that the effect was statistically equivalent to zero. When exploratorily testing equivalence for the partner effect we also observed that it was statistically equivalent to zero, while this was not the case for the oath effect (see Supplementary Fig. 5).

**Exploratory analyses.** Further, we tested for main and moderation effects of the specific designs of each study (e.g., different game tasks, number of rounds). A detailed overview can be found in the Supplementary Table 10–27. Overall, we found that participants behaved significantly more dishonestly when they could earn a bonus for themselves compared to earning one for another person from the study in Study 1, the mind shape task increased dishonesty significantly compared to the die roll task in Study 3 (but not in Study 2), and a one-shot task increased overall reports compared to four trials in Studies 4 and 5 but not statistically significantly[64]. We also explored the importance of different sociodemographic indicators (Supplementary Table 42–64 & Supplementary Figs. 13–15). We observed small effects of younger participants ($r = −0.08$ [−0.11, −0.05], $p < 0.001$) and men reporting higher scores ($d = 0.10$ [0.05, 0.15], $p < 0.001$) across the studies[7,34]. We also explored whether self-reported felt responsibility differed across treatments but observed no consistent evidence (Supplementary Figs. 11–12 and Supplementary Table 36–41).

We explored the moderation by Honesty-Humility and the development of dishonesty over time, which can be found in the Supplementary Figs. 7–10 and Supplementary Table 28–35. We did not observe any consistent effects. Overall, we observed a negative significant correlation between trait Honesty-Humility and reports in the game tasks ($r = −0.11$ [−0.15, −0.06], $p < 0.001$).

## Discussion
Across seven preregistered studies, three countries, and a total of 7566 participants, we investigated the effects of social

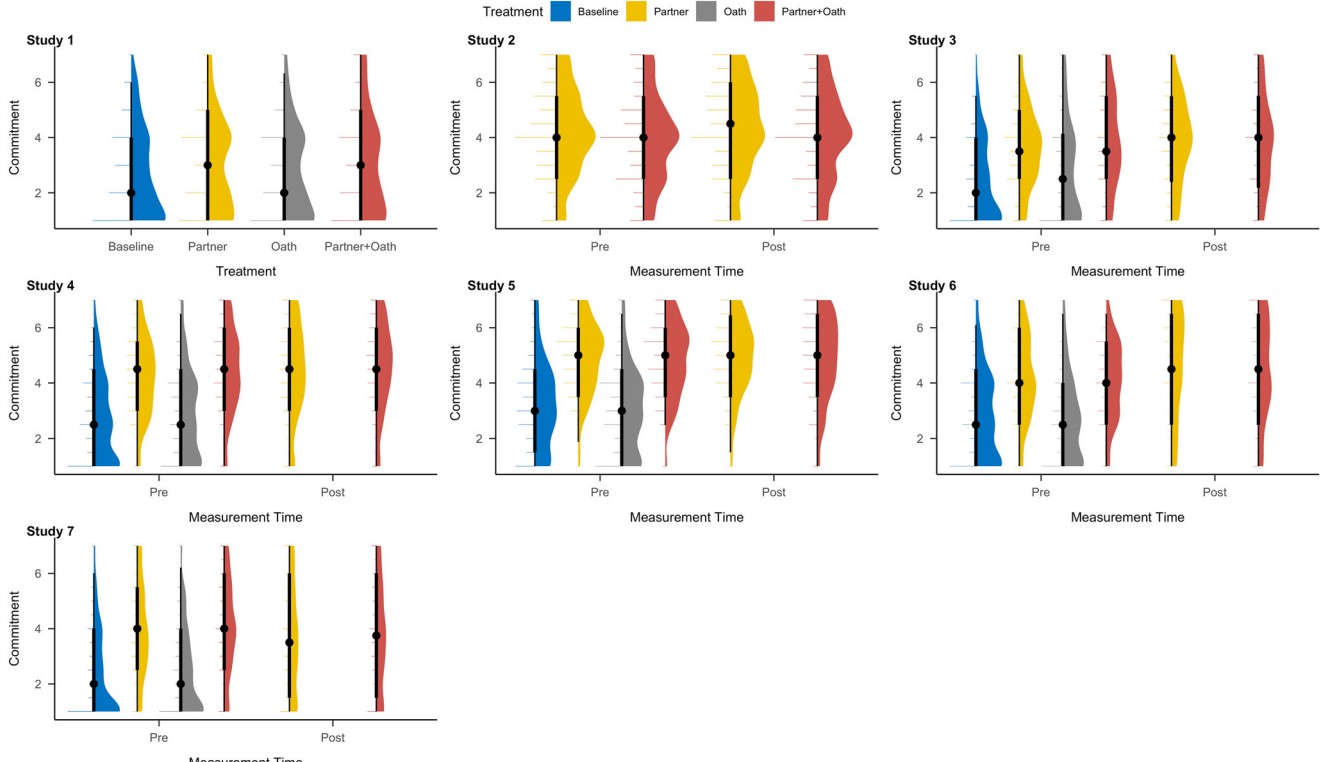

**Fig. 2 Commitment ratings to the partner or another person from the same study across the seven studies.** Raincloud plots for the commitment rating across the four treatments and seven studies. Each summary includes the distribution of the ratings, the boxplot, and individual data points. For the boxplot, the dot refers to the median, the thick lines to the interquartile range, and the thin lines to the minimum and maximum. Baseline treatment is depicted by the blue curve and data points, the partner treatment by yellow curves and data points, the oath treatment by grey curves, and the partner + oath treatment by red curves and data points. Note, that we did not assess commitment in the baseline and oath treatments in Study 2 due to a technical error. Independent samples: Study 1 (Baseline: $n = 198$, Partner: $n = 194$, Oath: $n = 188$, Partner + Oath: $n = 190$), Study 2 (Pre/Post Partner: $n = 373$; Pre/Post Partner + Oath: $n = 338$), Study 3 (Baseline: $n = 123$, Pre/Post Partner: $n = 135$, Oath: $n = 117$, Pre/Post Partner + Oath: $n = 109$), Study 4 (Baseline: $n = 432$, Pre/Post Partner: $n = 366$, Oath: $n = 406$, Pre/Post Partner + Oath: $n = 337$), Study 5 (Baseline: $n = 254$, Pre/Post Partner: $n = 231$, Oath: $n = 257$, Pre/Post Partner + Oath: $n = 240$), Study 6 (Baseline: $n = 237$, Pre/Post Partner: $n = 164$, Oath: $n = 191$, Pre/Post Partner + Oath: $n = 163$), Study 7 (Baseline: $n = 419$, Pre/Post Partner: $n = 345$, Oath: $n = 424$, Pre/Post Partner + Oath: $n = 352$).

commitment on dishonesty. We focused on commitment to individuals, commitment to moral norms, and, addressing a gap in the literature, the combined effect of both conditions.

**Commitment to other individuals.** Overall, we found no credible evidence that commitment to other individuals increased dishonest behavior, contrary to our prediction (H1) and to previous studies' findings[9,10]. Our tasks employed various methodologies that have successfully induced dishonesty among individuals in the past[9,42,43,62] including both no interaction and interaction among dyads, as well as repeated interaction or one-shot designs. The overall effect size ($\log(OR) = 0.08$, $d = 0.04$) was considerably small and much smaller than suggested by recent meta-analyses[4,7]. Noteworthy, one meta-analysis[4] observed increased evidence of publication bias, suggesting that the current effects might prove a more valid representation of the possible effects of commitment to other individuals on dishonesty. The fact that effects were similar across various designs including different commonly used manipulations, tasks, degrees of interaction, and populations, speaks to the robustness of our results. An exception was a stronger effect in Study 5 which focused on a Mexican population. An interpretation could be that the Mexican population was younger compared to the other samples. Previous meta-studies have provided evidence that younger people might be more likely to cheat (see Supplementary Fig. 15 for a replication of such age effects in the present studies). Recent meta-

studies showed a high reliance on US and Western European samples[4,7] calling for further cross-cultural work, especially since previous work has found mixed evidence regarding dishonesty across cultures[33,87]. Higher country levels of collectivism[88,89] or how cultures accept deviations from the norm[90] might influence the effects of social commitment on dishonesty. However, given the limited focus on cultural differences of the current studies, we are not able to provide strong evidence for this assumption.

Similarly, we found a somewhat stronger effect in the expected direction in Study 7 when employing a sequential die roll paradigm[9], though not statistically significant. As found in a recent meta-study[7], dishonesty might increase if the partner also acts dishonestly, which is more likely in a sequential interaction over several rounds such as in Study 7. We also found evidence that dyads that were more dishonest in the task reported higher commitment to their partner after the task, possibly reflecting justification in response to initial cheating. Nevertheless, the overall effect was small and considerably smaller compared to previous studies employing the sequential die roll paradigm[9,91]. This corroborates our overall finding of limited to no credible evidence that social commitment to other individuals increases dishonest behavior given the current manipulations, tasks, and contexts.

Across all studies, we observed a small positive correlation between commitment ratings and ratings in the economic game tasks (see Supplementary Figs. 16–19). Associations were slightly

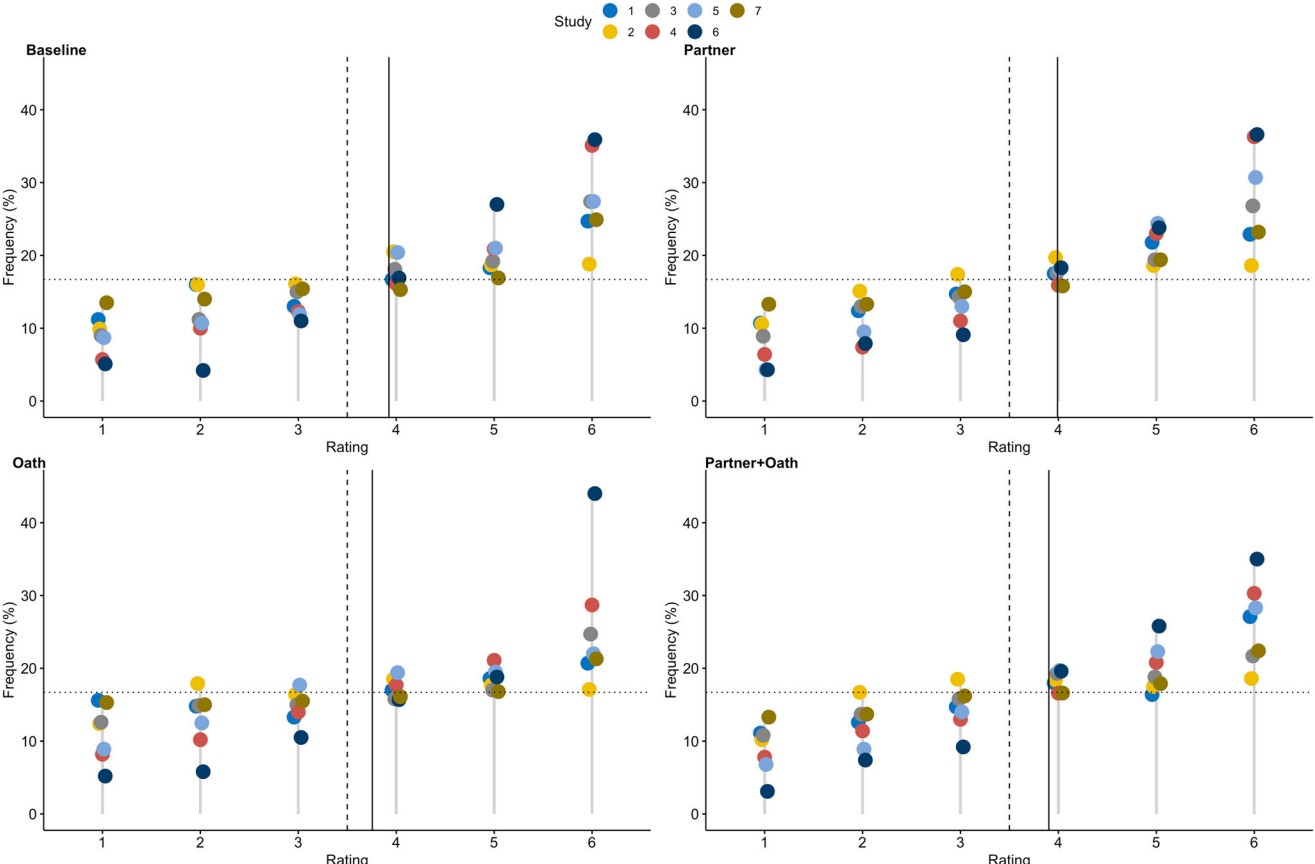

**Fig. 3 Distribution of economic game ratings across the seven studies and the four main conditions.** Cleveland dot plots illustrate the frequency (in %) of economic game ratings across the four treatments and seven studies. Ratings refer to the six different options in the dishonesty measures, whereby 1 refers to the lowest bonus (or a die roll of 1) and 6 refers to the highest possible bonus (or a die roll of 6). Dots refer to the frequency of the specific rating for the respective treatment and study. The blue dots depict Study 1, yellow dots depict Study 2, grey dots depict Study 3, red dots depict Study 4, light blue dots depict Study 5, dark blue dots depict Study 6, brown dots depict Study 7. Dashed vertical lines represent the expected mean if all participants would report honestly (3.5). Solid vertical lines represent the observed mean across all studies for the respective treatment. Dotted horizontal line represents the expected distribution if all participants report honestly (16.7%).

stronger for commitment ratings after the game, suggesting that social commitment might not only influence dishonesty but vice versa. It is possible that these relationships perpetuate and increase for repeated encounters[92].

**Commitment to moral norms.** As predicted (H2), we observed that committing to a moral norm by an honesty oath, reduced dishonest behavior to a degree that was small but not trivial, replicating previous studies[18,25,26,45,51–55]. Effects were comparable across different countries and dishonesty measures, except for Study 6. Although this finding was in the expected direction ($logOR = -0.23$ [$-0.31, -0.15$], $d = -0.13$), the effect size was considerably smaller than the effects reported in previous meta-analyses[4,5]. We employed a widely used form of oath as in prior studies[49], varying only the timing of the oath in Study 1 and small parts of the formulation in Study 7, which did not have an effect on (dis)honesty. It is possible that different forms of expressing moral norm commitment (e.g., signing instead of copying text, though see[27,93]) and norms (e.g., descriptive rather than injunctive norms) might produce differential and possibly stronger effects (see[94] for a framework to improve honesty nudges), but further studies are needed to confirm this effect. As previous studies did not find evidence for the effectiveness of commitment to oaths, especially in field settings[12,57,93], however, the present findings are reassuring. Nevertheless, it is important to study the

conditions under which such an intervention could reduce dishonesty. For instance, we observed that the oath increased dishonesty in Study 6. Such reactance to oaths has been observed in previous studies[95] and it would be interesting to investigate boundary conditions and the situational and personality variables that moderate such effects. A recent study suggested that oaths might work especially well for individuals who act honestly most of the time[26], but we found the smallest association between trait honesty and reporting for participants committing to the oath. Recent studies have started to investigate boundary conditions of honesty oaths[27,96], but more standardized procedures and formulations are needed to make different studies comparable and evaluate why and when honesty oaths can be (in)effective or to what degree. Future studies would also need to investigate more applied settings in which individuals commit to an organizational moral norm (e.g., code of conduct) instead of a general honesty norm.

**Commitment to other individuals and moral norms.** Finally, we found that feeling committed to another individual and a social honesty norm at the time did not have a credible effect on dishonest (or honest) behavior as predicted (H3). An interpretation could be that committing to a moral norm (such as via an honesty oath) is not as effective if one is at the same time also committed to another individual (compared to no felt commitment to other

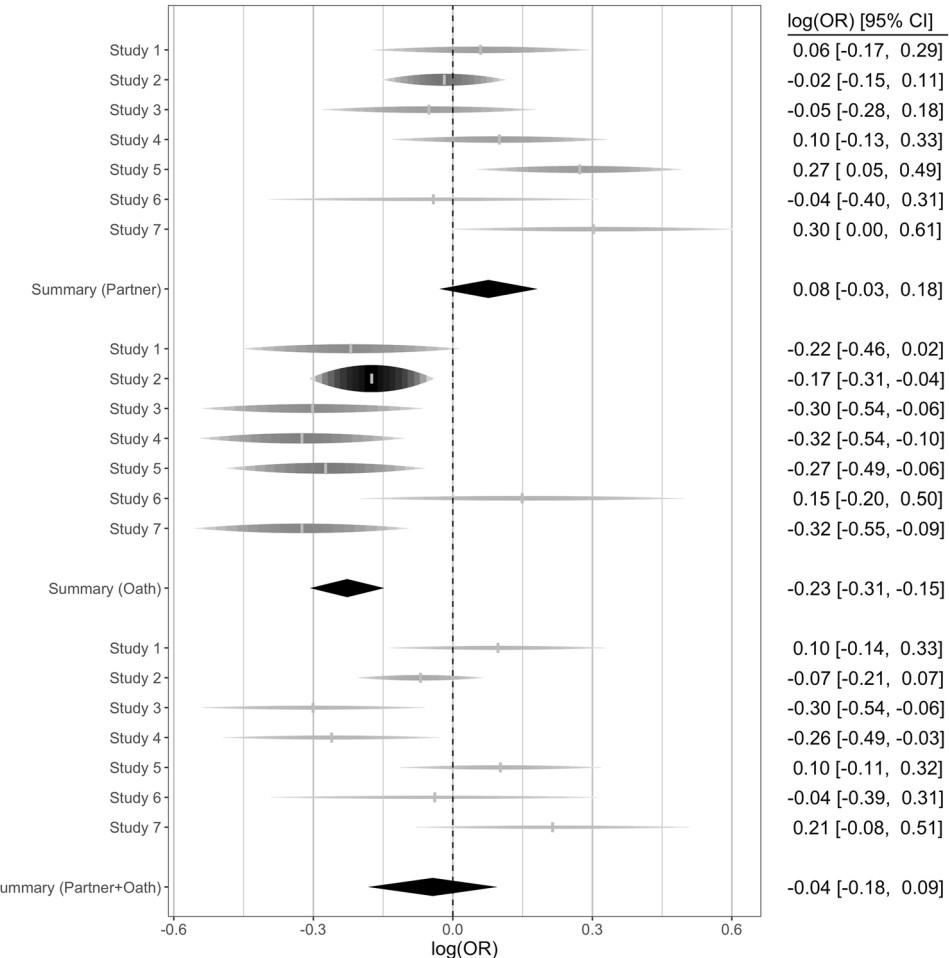

**Fig. 4 Forest plot of random effects meta-analysis by treatment compared to the baseline across the seven different studies.** Forest plot of individual effects for each experimental treatment compared to the baseline across the seven studies. Diamonds represent effect size in log Odds Ratios and 95% confidence intervals. The thickness of the diamonds indicates the weight of each study. Overall meta-analytical effects are presented as bold diamonds based on a random-effects model and restricted maximum likelihood estimation. The dashed vertical line represents log(OR) of 0 (indicating no change from baseline). Positive effects indicate an increase in ratings from the baseline (i.e., increase in dishonesty from baseline). Negative effects indicate a decrease in ratings compared to the baseline (i.e., decrease in honesty compared to the baseline). For Study 7, effect sizes are based on reported doubles thereby using one report per dyad. Independent samples (observations): Study 1 (Baseline: $n = 198$ (792), Partner: $n = 194$ (776), Oath: $n = 188$ (752), Partner + Oath: $n = 190$ (760)), Study 2 (Baseline: $n = 415$ (1660), Partner: $n = 373$ (1492), Oath: $n = 368$ (1472), Partner+Oath: $n = 338$ (1352)), Study 3 (Baseline: $n = 123$ (1230), Partner: $n = 135$ (1350), Oath: $n = 117$ (1170), Partner + Oath: $n = 109$ (1090)), Study 4 (Baseline: $n = 432$ (1095), Partner: $n = 366$ (843), Oath: $n = 406$ (1084), Partner + Oath: $n = 337$ (874)), Study 5 (Baseline: $n = 254$ (647), Partner: $n = 231$ (579), Oath: $n = 257$ (650), Partner + Oath: $n = 240$ (615)), Study 6 (Baseline: $n = 237$, Partner: $n = 164$, Oath: $n = 191$, Partner+Oath: $n = 163$), Study 7 (Baseline: $n = 419$ (4190), Partner: $n = 167$ (1670), Oath: $n = 424$ (4240), Partner+Oath: $n = 180$ (1800)).

individuals). We found across studies that committing to an honesty norm while also committing to another individual reduced scores in contrast to committing to another individual without the honesty norm, but increased scores when comparing it to committing to an honesty norm without any other commitments. Practically, this seems to suggest that interventions such as oaths are not as effective in reducing dishonest behavior when one's actions are dependent on and benefiting other individuals. The good news is that employing an oath in such a setting still reduced dishonest behavior to a small degree (in contrast to no oath) and does not seem to have any negative (i.e., backfiring) effects, as might be suggested by prior work on autonomy motives and reactance[95,97,98].

Is it problematic that we did not find a credible effect of feeling a commitment to other individuals to interpret the combined effect of social commitment? Looking at studies 5 and 7, in which commitment to other individuals increased dishonesty

significantly, indicate that adding an honesty oath to the same design decreases the occurrence of dishonesty. This suggests that overall, morality and prosociality concerns might neutralize each other. This can be further contextualized by focusing on Study 7 in which participants in the partner and partner + oath treatments were faced with different prosociality concerns based on their role in the sequential die roll task. When taking the role of Player A, there exists some pressure to provide higher ratings, but this is not crucial for the group to succeed. On the other hand, when taking the role of Player B, there is a high prosocial concern about matching the partner's report for the group to succeed. In fact, our results demonstrate that dishonesty was higher, rather than lower, for Player B who also committed to an honesty oath. This corroborates unpublished data suggesting that commitment to moral norms is less effective when the outcomes of a group fully depend on the other group members[19], possibly because prosocial norms are more important than moral norms[7]. Future

studies would need to test this assertion more systematically and investigate boundary conditions when one norm might override the other.

**Limitations**. Although we employed a large variation of designs, tasks, payoffs, and populations there are still some limitations to note in the current studies. First, assessing the effectiveness of the commitment to other individuals was difficult, as it was not obvious which reference group should be used in the conditions in which no actual partner was involved. We also failed to assess felt commitment to the moral norm throughout the studies. We theorized that felt responsibility for one's action might be a common mechanism of social commitment to other individuals and moral norms, with felt responsibility increasing with more commitment to the moral norm and decreasing with more commitment to other individuals. However, we failed to find any consistent effects. Future studies would need to assess commitment to both other individuals and moral norms more systematically to investigate possible mechanisms of the obtained effects. Relatedly, while our theoretical model assumes that participants felt social commitment to the experimenter when completing the oath, this was not explicitly mentioned. Future studies would need to make this aspect more salient, which should increase social commitment based on the theoretical assumption.

Second, the effect of our manipulation was also rather small in Study 1. We tried to address such limitations by varying the manipulation of commitment across the seven studies. Our studies also call into question whether collaborative corruption exists in an online context. However, previous studies have found much stronger effects when employing paradigms with no or a similar amount of minimal interaction as in the current studies[9,10]. It is likely that stronger effects might exist for stronger and more detailed manipulations that focus on inducing commitment in repeated or real-life interactions.

Third, dishonesty rates differed strongly across the different tasks and populations[64]. It is essential to encourage dishonest behavior to be able to study it, which is why we employ private instead of public or traceable tasks. Some baseline dishonesty rates were rather low (e.g., $M = 3.72$ in Study 2), which might have influenced the effectiveness of our manipulations, particularly the honesty oath. Notwithstanding, an oath did significantly decrease dishonest behavior. This finding might also be related to the specific crowdsourcing populations we used. Participants, especially panelists in Study 2, might fear getting rejected for behaving dishonestly or being removed from the panel and not able to participate in future studies[99]. Studies would need to highlight the absence of punishment or study dishonesty in settings where fear of punishment is reduced as much as possible, while at the same time avoiding demand characteristics. The majority of studies were conducted on Prolific.co which has been considered to provide good data quality[100]. On the other hand, these participants are also homogenous to a certain degree and have potentially taken part in numerous studies using similar paradigms or might act dishonestly because they think the experimenter wants them to[101].

Fourth, due to the design of the studies we were unable to fully randomize the order of the different manipulations. In Studies 1–3 and 6–7 participants always committed to the honesty oath before being presented with the commitment to partner manipulation. In Study 4 and 5 parts of the commitment to partner manipulation were presented before the honesty oath. A fully randomized design was difficult due to the fact that for Studies 2–7 the commitment to partner manipulation was part of the main task that also measured our dependent variable. This should be considered when interpreting the effect of both commitments.

## Conclusion

Employing seven high-powered preregistered studies, we found no credible evidence that feeling committed to other individuals increases dishonesty. The overall effect was too trivial to be of practical importance suggesting that there is no credible evidence for corrupted collaboration, at least given the current methods and in the given settings. On the other hand, commitment to moral norms via honesty oaths might be effective in reducing dishonest behavior, although the effect was small as well. Our studies provide some practical grounding when it comes to factors influencing dishonesty (i.e., asking people to express written commitment to behave honestly might help in curbing dishonesty, especially in non-collaborative contexts). Nevertheless, they also suggest that effect sizes in dishonesty research and moral psychology are likely smaller than what has been reported during the last decades, mirroring recent developments across several disciplines of psychology.

## Data availability

The datasets[102] generated by the survey research during and/or analyzed during the current study are available in the osf.io repository, https://osf.io/sxbfn/. Additional data including partial post code data and chat logs are not included in the open repository due to privacy regulations. These data can be available upon request.

## Code availability

All code[103] used to generate the main and supplemental analyses is available in the osf.io repository, https://osf.io/sxbfn/. For all analyses we used R (version 4.0.3[104]) and the following packages: metafor (version 3.0-2[86]), ordinal (version 2019.12-10[85]), MASS (version 7.3-53[105]), tidyverse (version 1.3.0[106]), ggpubr (version 0.4.0[107]), TOSTER (version 0.3.4[108]), simr (version 1.0.6[109]), lme4 (version 1.1-32[110]).

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

## Acknowledgements

The project was funded by the Aarhus University Foundation (AUFF), Nova Grant: AUFF-E-2019-9-4. The funders had no role in study design, data collection, and analysis, the decision to publish, or the preparation of the manuscript. We thank Ana Sofía Ramirez Gonzalez for her help with setting up Study 5 and Andras Molnar for his help with the SMARTRIQS application. The current studies were conducted after a first version of the meta-analysis (Zickfeld et al., 2022) was submitted for publication. During the revision round of the meta-analysis all studies that had been conducted at that time (Study 1-6) were added to the meta-analysis based on an updated search. Study 7 was conducted after the meta-analysis was resubmitted for publication and later included in a revised version.

## Author contributions

J.H.Z.: Conceptualization, Methodology, Formal Analysis, Investigation, Data Curation, Writing – Original Draft. K.A.Ś.: Conceptualization, Methodology, Writing – Review & Editing. A.W.: Conceptualization, Methodology, Writing – Review & Editing; J.M.: Conceptualization, Methodology, Writing – Review & Editing, Funding Acquisition. P.M.: Conceptualization, Methodology, Writing – Review & Editing, Funding Acquisition.

## Competing interests

The authors declare no competing interests.
