## [Peer Review File · Communications Psychology]

19th Dec 22

Dear Dr Zickfeld,

Thank you for your patience during the peer-review process. I am sorry that the decision comes with a slight delay. Your manuscript titled "Investigating the Impact of Social Commitment on Dishonest Behavior" has now been seen by 3 reviewers, whose comments are appended below. You will see that they find your work of some potential interest. However, they have raised quite substantial concerns that must be addressed. In light of these comments, we cannot accept the manuscript for publication, but would be interested in considering a revised version that fully addresses these serious concerns.

We hope you will find the Reviewers' comments useful as you decide how to proceed. Should additional work allow you to address these criticisms, we would be happy to look at a substantially revised manuscript. If you choose to take up this option, please highlight all changes in the manuscript text file, and provide a detailed point-by-point reply to the reviewers.

Editorially, we consider three issues key for a revision (listed below). As you address these issues and the referees other remarks, please bear in mind the journal's requirements for statistics interpretation and reporting, as well as our preregistration reporting requirements. Further details on general journal standards are at the end of this letter.

1) You will find that the referees express doubt as to whether the results provide a conclusive answer to H3, which may partly a design issue. We ask you to address this point in revision, providing further empirical data to support your interpretation. For all null-results reporting, please adhere to the journal standards for statistics reporting and interpretation as detailed here: <https://www.nature.com/commspsychol/submit/submission-guidelines#statistical-guidelines> ; in brief, null-results may only be interpreted on the basis of Bayesian tests or equivalence tests (positive evidence for the null).

2) The referees express concerns about the theoretical framing and how the design (and predictions) map onto the theory, or onto previous work. We ask you to clarify the framework and how you derive predictions, as well as explain how the design allows testing these predictions. Remaining ambiguities should be discussed as caveats in the discussion section. Please note that we do not ask you to reframe the paper or change the hypotheses. Please also note that the order of sections in published Articles in Communications Psychology runs as follows: Abstract, Introduction, Methods, Results, Discussion. We encourage you to adopt this order of sections now, as you address the referees' concerns.

3) Your work was preregistered, which is commendable. Please note our guidelines for preregistration reporting, which you can find here: <https://www.nature.com/commspsychol/submit/preregistration> ; please address Reviewer #2's concerns about differences between preregistration and actual sample. Tangentially related, we ask you to refrain from providing a post-hoc power analysis, and instead report a sensitivity analysis, as

described in Lakens, D. (2022). Sample size justification. *Collabra: Psychology*, 8(1), 33267.

If the revision process takes significantly longer than five months, we will be happy to reconsider your paper at a later date, provided it still presents a significant contribution to the literature at that stage.

We understand that due to the current global situation, the time required for revision may be longer than usual. We would appreciate it if you could keep us informed about an estimated timescale for resubmission, to facilitate our planning. Of course, if you are unable to estimate, we are happy to accommodate necessary extensions nevertheless.

Please use the following link to submit your revised manuscript, point-by-point response to the Reviewers' comments with a list of your changes to the manuscript text (which should be in a separate document to any cover letter) and any completed checklist:

[link redacted]

Please do not hesitate to contact me if you have any questions or would like to discuss the required revisions further. Thank you for the opportunity to review your work.

Best regards,

Marike

Marike Schiffer, PhD
Chief Editor
Communications Psychology

EDITORIAL POLICIES AND FORMATTING

Editorial Policy: [Policy requirements](https://www.nature.com/documents/nr-editorial-policy-checklist.pdf) (Download the link to your computer as a PDF.)

Furthermore, please align your manuscript with our format requirements, which are summarized on the following checklist:

<https://www.nature.com/documents/commsj-psychol-style-formatting-checklist-article.pdf>>Communications Psychology formatting checklist

and also in our style and formatting guide <https://www.nature.com/documents/commsj-psychol-style-formatting-guide-accept.pdf>>Communications Psychology formatting guide .

* **CODE AVAILABILITY:** All Communications Psychology manuscripts must include a section titled "Code Availability" at the end of the methods section. In the event of publication, we require that the custom analysis code supporting your conclusions is made available in a publicly accessible repository; please choose a repository that provides a DOI for the code; the link to the repository and the DOI must be included in the Code Availability statement. Publication as Supplementary Information will not suffice. We ask you to prepare and upload code at this stage, to avoid delays later on in the process.

* **DATA AVAILABILITY:**

All Communications Psychology research manuscripts must include a section titled "Data Availability" at the end of the Methods section or main text (if no Methods). More information on this policy, is available at <http://www.nature.com/authors/policies/data/data-availability-statements-data-citations.pdf>><http://www.nature.com/authors/policies/data/data-availability-statements-data-citations.pdf>.

At a minimum the Data availability statement must explain how the data can be obtained and whether there are any restrictions on data sharing. Communications Psychology strongly endorses open sharing of data. If you do make your data openly available, please include in the statement:

We recommend submitting the data to discipline-specific, community-recognized repositories, where possible and a list of recommended repositories is provided at <http://www.nature.com/sdata/policies/repositories>><http://www.nature.com/sdata/policies/repositories>.

If a community resource is unavailable, data can be submitted to generalist repositories such as <https://figshare.com/>>figshare or <http://datadryad.org/>>Dryad Digital Repository. Please provide a unique identifier for the data (for example a DOI or a permanent URL) in the data availability statement, if possible. If the repository does not provide identifiers, we

encourage authors to supply the search terms that will return the data. For data that have been obtained from publicly available sources, please provide a URL and the specific data product name in the data availability statement. Data with a DOI should be further cited in the methods reference section.

REVIEWERS' EXPERTISE:

All reviewers share expertise in: social psychology, honesty/dishonesty, economic games, minimal group designs

REVIEWERS' COMMENTS:

Reviewer #1 (Remarks to the Author):

The authors conducted six highly powered studies to examine how commitment to other individuals vs groups may cast different levels of deterrence on dishonesty. I started reading the paper with high expectations and great interest. Indeed, a deeper and more systematic understanding of commitment and dishonesty can help elucidate the social, economical, and psychological effects of a practice that has steadily grown into a part of our organizational norm. I also applaud the authors' effort to include actual interactions between participants online. Ultimately, however, I found the paper to fall short on several fronts, which I expand on below. I hope the authors will find these comments to be useful as they further improve this paper. Where is the theory? This paper aims to tackle an important question that holds the potential of bearing both theoretical and practical implications above and beyond what we already know from the literature. However, I found the theory portion of this paper to be quite superficial and thin. For example, the authors neglected to provide a concrete definition and operationalization of dishonesty. Likewise, the authors did not provide a concrete definition of social norms. This led to a mismatch between the working definition of social commitment to the norm in the introduction section and the actual operationalization of norms in the six studies that the authors conducted. I encourage the authors to tighten their key constructs. This will inevitably require the authors to conduct a deeper and more comprehensive literature review.

What is the context? As it stands, the results section is very difficult for readers to understand. It felt as if the authors wrote a paper aimed at a more disciplinary journal, then simply moved the methods section to the back to fit the requirements of the current journal. I understand that the authors should be in control of how they structure and write their paper. However, I highly recommend a complete re-write of the current paper.

How should I make sense of the null hypothesis (H3)? I agree with the authors that, of the three key hypotheses, H3 is by far the most interesting. However, I am not sure how to interpret null hypothesis. Is it that the experiment did not work? Or is it that the two forces truly canceled out. In terms of this paper, the result showed inconsistent results for commitment to partners, and null effect for both partner and oath. Therefore, I do not think that the current paper provides sufficient evidence for the authors to claim that H3 is supported.

Methods:

Why those three countries? These three countries differ in terms of collectivism, with Mexico being the most collectivistic out of the three. How would that affect the strength of participants' commitment to their partners? Similarly, these three countries also differ in terms of their cultural tightness, with US being the loose and accepting of deviations from the norm. How would the authors

I appreciate the detailed description of power analysis, sample size determination, data recruitment process. I encouraged the authors to reallocate their effort and put in the same level of attention to describing the actual procedures of each study.

The research seems to be tilting in favor of social norms. On the one hand, they are pledging to a norm of honesty. On the other hand, they are pledging to the wellbeing of a person. A better and cleaner comparison would be to examine the effect of pledging to the org culture/norm, instead of norm of honesty. i.e., Hippocratic oath.

Why not randomize the order of partner and oath commitment manipulations? I did see note 12, which I assumed was the author's justification for not randomizing the conditions. However, that leaves the possibility that those in the partner + oath condition could have been anchored by the partner manipulation. In turn, this contributed to the null interaction effect.

I found the mind shape task to be quite clever and novel. However, as the authors indicated, the task did not allow the authors to capture the actual degree of cheating. As the authors move forward with this research, I encourage the authors to include at least one study that captures cheating in action.

Minor notes:

p.4: Commitment to other individuals: within or outside of the organization? Might want to clarify Beneficiary matters? Dishonesty to benefit the self vs. benefit the group. See Emma Levine's work for more on these distinctions.

Reviewer #2 (Remarks to the Author):

Report on "Investigating the Impact of Social Commitment on Dishonest Behavior"

The authors try to replicate previous studies investigating the impact of social commitment (feeling committed to other persons) and commitment to social norms on dishonest behavior. They use two manipulations of social commitment (minimal group paradigm and live interaction) and an honesty oath to manipulate commitment to the social norm of telling the truth. Furthermore, they use two experimental tasks measuring the dependent variable lying behavior (mind game and die-in-the-cup task). The authors add to the previous literature the combination of social commitment and commitment to social norms. With more than 6,000 participants in pre-registered experiments, they find evidence that honesty oaths decrease dishonesty. They cannot replicate that commitment to other persons increases dishonesty. Finally, the authors observe a null effect when combining social commitment and commitment to social norms.

Given the replicability and credibility crisis in experimental psychology, more of these studies should be published. In my view, this applies also to the current study after the comments below are considered.

The limitations of the current study are no measurement of commitment in the baseline and oath treatments of Study 2, the manipulation checks of social commitment to other persons (in general), and the large variance of dishonesty across subject pools. However, the latter two limitations also apply to previous studies, and the limitations are mentioned.

Specific comments

1. In my view, the authors should elaborate a bit more on the limitations, especially the latter two. Furthermore, the pros and cons of using Prolific could be discussed.
2. I think the authors should more clearly disclose and discuss the relationship of the current study to the meta-analysis of Zickfeld et al. (2022).
3. Given their results, the authors should discuss in more detail the role of honesty oaths and how their design and findings contribute to previous literature. Some studies on honesty oaths are considered in Zickfeld et al. (2022) but not addressed in the current paper (e.g., Beck et al., 2020).
4. I think the graphs and tables need a better description and explanation in the text.
5. I expected the Method section before the Results section.
6. Although pre-registered, I was wondering about the author's approach to stop the recruiting once the main effects were statistically significant. I think this could be discussed in the paper.
7. The suggested numbers of independent observations (to achieve a power of 0.8) from the ex-ante power analyses differ substantially from the planned and finally realized numbers of independent observations – Study 1: 1060 vs. 770, Study 2: 2320 vs. 1494, Study 3: 7408 vs. 484, Study 4: 2160 vs. 1541, Study 5: 2160 vs. 982, Study 6: 1840 vs. 755. When justifying the sample size in the supplementary material, the authors seem to mix up the number of data points (observations) and the sample size (number of independent observations). What is the achieved power in the studies?

Reviewer #3 (Remarks to the Author):

The present work addresses how commitment to other individuals and to social norms influences dishonest behavior. In a set of six studies using different kinds of incentivized cheating paradigms, the authors systematically manipulated whether dishonest behavior affected another person's outcome (social commitment: yes vs. no) and whether participants had to commit themselves to an oath (norm commitment: yes vs. no). Across studies, they found that commitment to others did not affect dishonesty, whereas commitment to an oath reduced dishonest behavior. In turn, commitment to an oath was less effective in decreasing dishonesty if participants were at the same time socially committed to another person.

Overall, I very much enjoyed reading this paper. It investigates an important and timely issue, it is well-written, and the methodology is sound. Moreover, I want to applaud the authors for their strong commitment to open science, having pre-registered all studies and providing all materials,

data, and analysis scripts online. That said, however, I do have concerns about the framing of the paper and the general contribution. Specifically, I don't think that the two factors the authors investigate should be considered instances of the same variable, that is, social commitment. Rather, the main contribution of the present paper seems to be that the authors are the first to combine two variables that have been shown to influence (dis)honest behavior. I detail these and a few additional concerns in what follows. In general, I don't think that these issues can't be healed in a revision, but it would probably require substantial rewriting.

1) The way how commitment is defined in the abstract (i.e., "feeling committed to other individuals or groups") and on the very first page of the manuscript (i.e., "motivation to perform specific actions other agents are relying on") does not fit the manipulation of social commitment via social norm compliance. Specifically, in the norm compliance condition, participants were asked to commit to an oath reading as follows: "Participants in this study commit to the norm of telling the truth. I promise that the information I am providing in this study is true". Thus, in this condition, there is no other individual (or group) involved to which one can commit oneself – it is only about whether people commit to tell the truth, which may rather be considered a moral norm that is defined by the experimenters than a generally held social norm. In this regard, I was also wondering about the "social commitment to other individuals" conditions. Here, dishonest behavior profits another participant by increasing their (monetary) outcome. Thus, dishonest behavior is prosocial, which is why one could likewise consider this manipulation to touch upon a social norm: the norm of prosociality. So, in general, I found the framing somewhat misleading and kept wondering about what exactly the authors manipulated here and what the studies can tell us. In my view, the major contribution of the present paper is that commitment to an oath is combined with manipulating the social justifiability of lying. This, however, would require a different framing of the manuscript and a major overhaul of the reasoning in general.

2) I think the surprisingly weak effect observed for commitment to another individual may be attributable to the specific manipulation of social commitment used. Specifically, participants knew that their own and the other's individual outcomes would be summed up and then distributed among the two. Thus, both acted independently of each other and both their behavior influenced each person's outcome. Arguably, previous research showing an effect of commitment to others on dishonesty used stronger manipulations, for example, by having participants work together to increase their joint outcome (i.e., collaborative corruption) or by having another party profit from the individual's dishonesty while this other party had no opportunity to obtain a reward themselves (i.e., they did not participate in the cheating task). In the present studies, the felt obligation to help the other (i.e., the partner) was thus likely lower than in previous studies using different manipulations. Is there any evidence that the authors can refer to to test this idea or show the comparability of their manipulation of social commitment with other manipulations? This would greatly help assess the conclusiveness of the findings at hand.

3) Hypothesis 3 suggests that the effects of the two manipulated factors are similar in size. Is this assumption really justified based on previous evidence? I think more needs to be done to justify that combining the two manipulations should indeed yield a zero effect because they will cancel each other out.

More minor comments:

4) It would be useful to provide at least a brief summary of the methods at the end of the introduction. Otherwise, it is very difficult to follow the remainder of the paper without reading the

Methods section first.

5) It should be specified that Figure 2 refers to commitment to the partner, rather than commitment to the oath.

6) The abbreviation "IOS" (in the Design & Procedure section) is not defined.

Rebuttal Letter

*Original comments are provided in Calibri and our responses in **BOLD**.*

R1.

The authors conducted six highly powered studies to examine how commitment to other individuals vs groups may cast different levels of deterrence on dishonesty. I started reading the paper with high expectations and great interest. Indeed, a deeper and more systematic understanding of commitment and dishonesty can help elucidate the social, economical, and psychological effects of a practice that has steadily grown into a part of our organizational norm. I also applaud the authors' effort to include actual interactions between participants online. Ultimately, however, I found the paper to fall short on several fronts, which I expand on below. I hope the authors will find these comments to be useful as they further improve this paper.

Where is the theory? This paper aims to tackle an important question that holds the potential of bearing both theoretical and practical implications above and beyond what we already know from the literature. However, I found the theory portion of this paper to be quite superficial and thin. For example, the authors neglected to provide a concrete definition and operationalization of dishonesty. Likewise, the authors did not provide a concrete definition of social norms. This led to a mismatch between the working definition of social commitment to the norm in the introduction section and the actual operationalization of norms in the six studies that the authors conducted. I encourage the authors to tighten their key constructs. This will inevitably require the authors to conduct a deeper and more comprehensive literature review.

We have provided a more detailed theoretical review of the literature and how we define and operationalize our main concepts.

We have provided a more detailed definition of dishonesty and (social) norms. We explicitly define dishonesty as “intentionally misreporting private information (Feess & Kerzenmacher, 2018)” (p. 7) and (moral) norms as a “shared understanding of what is right or wrong in a given context and community (Ostrom, 2000)” (p. 3) and “other individuals in a given community or society demand of each other to follow it and do so with a certain prevalence (see Malle, 2023).” (p. 7)

Further, we included a section to define our approach to social commitment and how both types of commitment can be considered as a *social* commitment.

We adopt a minimal definition of (social) commitment, arguing that:

“[...] we adopt a minimal definition of social commitment arguing that social commitment refers to the dispositional state of an agent (X), who is motivated to carry out an action (z) because some other agent(s) (Y) is relying on them to do so (Michael et al., 2016; Zickfeld et al., 2022). Thus, social commitment can refer to a specific action or a general motivational dispositional state of agent X in relationship to agent Y. It is also sufficient

that X thinks that Y is relying on the specific action. In contrast to self-commitment (e.g., committing to the goal of eating less candy), social commitment always involves at least two agents. Social commitment can further be described as unilateral, only one agent feeling committed to the other, and mutual, both agents feeling committed to each other by the same (joint) or different (complementary) goal (Clark, 2020).“ (p. 5)

Further, we also provide a theoretical consideration how social commitment can affect dishonesty, by arguing that commitment to moral norms and individuals makes different norms salient (telling the truth vs. cooperation/loyalty) and that this balance decides whether the individual will act dishonestly or not (Weisel & Shalvi, 2022).

“Considering all the previously mentioned points, why would different forms of social commitment result in various outcomes with regard to honest behavior? The two types of commitment are likely to make two different norms salient: a moral concern for honesty (i.e., wanting to be an honest person; Hertwig & Mazar, 2022) and a prosocial and loyalty concern targeted specifically at the other individual (i.e, wanting to cooperate with the other individual(s) and wanting to behave loyal to the other individual(s), Fehr & Fischbacher, 2004). When the concern for cooperation collides with the concern for honesty, individuals might be willing to trade their moral currencies, and prioritize prosocial concerns, in order to justify their dishonest behavior, especially if they can gain from it (Weisel & Shalvi, 2022). We expect that stronger commitment to other individuals should increase attendance to prosociality norms and concerns whereas, stronger commitment to a general moral norm (e.g., telling the truth) should increase attendance to moral norms and concerns. Similarly, it is likely that social commitment to individuals and moral norms affects felt responsibility for one's actions differently. Whereas feeling committed to moral norms should increase felt responsibility, this should be reduced for commitment to other individuals.” (pp. 9-10)

What is the context? As it stands, the results section is very difficult for readers to understand. It felt as if the authors wrote a paper aimed at a more disciplinary journal, then simply moved the methods section to the back to fit the requirements of the current journal. I understand that the authors should be in control of how they structure and write their paper. However, I highly recommend a complete re-write of the current paper.

We acknowledge that this was a misunderstanding, as the article was originally submitted to a different journal and automatically transferred to the current journal and we failed to update the specific section order.

In line with the guidelines of the current journal we have now moved the Method section before the Results section and hope that this facilitates understanding considerably.

How should I make sense of the null hypothesis (H3)? I agree with the authors that, of the three key hypotheses, H3 is by far the most interesting. However, I am not sure how to interpret null hypothesis. Is it that the experiment did not work? Or is it that the two forces truly canceled out. In terms of this paper, the result showed inconsistent results for commitment to partners, and null effect for both partner and oath. Therefore, I do not think

that the current paper provides sufficient evidence for the authors to claim that H3 is supported.

We have now improved our theoretical and empirical argument why we predict a null effect for H3. We argue that commitment to moral norms and individuals at the same time activates concerns to both honesty and prosocial norms, which cancel each other out. Based on findings from a recent meta-analysis (Zickfeld et al., 2022), findings were indeed comparable in opposite directions and we predicted that these will cancel each other out. We now write:

“From a theoretical perspective, we would expect that social commitment to other individuals and moral norms at the same time, induces attendance to both prosocial norms on the one hand (i.e., wanting to help the other individual or the group) and attendance to moral norms on the other hand (i.e., wanting to be an honest person). Whether one of these can override the other might depend on the salience of each concern or norm (i.e., the strength of the (social) commitment). In the Zickfeld and colleagues (2022) meta-analysis both effects were similar in strength ($g = -.17$ vs. $g = .24$), suggesting that combining both commitments might cancel each other out. Therefore, we predict that:

H3. Combining social commitment to individuals and to an honesty oath (compared to the baseline) does neither increase nor decrease dishonest behavior (null effect).” (p. 11)

In addition, we think that our fully crossed 2x2 design allows us to test whether H3 is supported or not. Our hypothesis states that we expect no credible difference between the treatment that includes both types of commitment (commitment to honesty oath and commitment to partner) and the treatment where both of these manipulations are absent. We define a smallest effect size of interest ($d = .15$) and perform an equivalence test that gives us evidence that the effect of the combined treatment is significantly smaller than this smallest effect size of interest. Therefore, H3 is supported as such.

Now, you can argue and question whether effects cancel each other out if there is no credible effect for commitment to a partner only in the first place. Indeed we find no overall credible effect of commitment to a partner (compared to the baseline, although it is in the expected direction). Nevertheless, the effect is stronger than when combining both manipulations (logOR: $-.04$ for the combined vs. logOR: $.08$). Testing both treatments against each other directly shows evidence that the commitment to moral norm/individual treatment effect on dishonesty is significantly smaller than the commitment to individual effect, and significantly higher than that for the commitment to moral norm only treatments. Therefore, we can argue that introducing an honesty oath in a group setting can reduce dishonesty (in comparison to the group setting) but does not work as well as compared to using an honesty oath in an individual setting.

Therefore, we believe that the results provide support for H3.

Methods:

Why those three countries? These three countries differ in terms of collectivism, with Mexico being the most collectivistic out of the three. How would that affect the strength of

participants' commitment to their partners? Similarly, these three countries also differ in terms of their cultural tightness, with US being the loose and accepting of deviations from the norm. How would the authors

Unfortunately, the comment was cut off in the middle of the last sentence and so we are not able to comment on the reviewer's concluding question.

We have now made clear that we focus on these three countries for reasons of convenience and to test the generalizability of our effects.

"We selected the three populations to investigate the generalizability of our effects. The three countries were chosen out of convenience. More detailed information on their differences is provided in Supplement Section 20." (footnote 6)

These populations differ indeed on several country-level indicators as mentioned and we have provided a detailed overview in Supplement Section 20. As mentioned, Mexico is the most collectivistic out of the three countries, which might be related to the fact that they show the strongest effect when feeling committed to another individual (Mazar & Aggarwal, 2011; Jamaluddin et al., 2021). Further Mexico is the *tightest* country in terms of cultural tightness which has been argued to increase high-levels of dishonesty (Aycinena et al., 2022). However, with three countries and only using the same design in two of these countries we feel that it is difficult to make any assertions about cultural differences. What seems most reassuring to us is the fact that the effect of commitment to an honesty oath was similar across the different samples (except in Study 6). We added a short paragraph to reflect this:

"Higher country levels of collectivism (Mazar & Aggarwal, 2011; Jamaluddin et al., 2021) or how cultures accept deviations from the norm (Aycinena et al., 2022) might influence the effects of social commitment on dishonesty. However, given the limited focus on cultural differences of the current studies we are not able to provide strong evidence for this assumption. " (p. 38)

I appreciate the detailed description of power analysis, sample size determination, data recruitment process. I encouraged the authors to reallocate their effort and put in the same level of attention to describing the actual procedures of each study.

We think that moving the Method section before the Results section has considerably improved the understanding of the methods and procedures.

We agree that a focus on describing the procedures and designs of each study meticulously is important for replication purposes and understanding strength and weaknesses of the studies. However, given the focus on seven studies and page constraints on the other hand, we think that our description is sufficient to get an idea of the main procedures. A detailed description of each study and its procedures is provided in the Supplementary Material (Section 1.1-1.6) and we also provide all materials on our osf.io project page.

The research seems to be tilting in favor of social norms. On the one hand, they are pledging to a norm of honesty. On the other hand, they are pledging to the wellbeing of a person. A better and cleaner comparison would be to examine the effect of pledging to the org culture/norm, instead of norm of honesty. i.e., Hippocratic oath.

We have revised our formulation and now refer to *commitment to moral norms* and not social norms (also in response to R3). We agree that it would be interesting to compare commitment to another person (within an organization) and commitment to the organizational norm on the other hand. However, this is not the specific focus of the current paper. We have added a sentence to acknowledge the importance of studying this aspect in future studies:

“Future studies would also need to investigate more applied settings in which individuals commit to an organizational moral norm (e.g., code of conduct) instead of a general honesty norm.” (p. 39)

Why not randomize the order of partner and oath commitment manipulations? I did see note 12, which I assumed was the author’s justification for not randomizing the conditions. However, that leaves the possibility that those in the partner + oath condition could have been anchored by the partner manipulation. In turn, this contributed to the null interaction effect.

We agree that it would be the cleanest design to randomize the order of the different commitment manipulations. However, given our designs this would have only been possible in Study 1 as for the other studies the commitment to partner manipulation is the same task as the dependent variable (dishonesty measure). We have added this as a limitation of the current studies.

We were unsure what the reviewer meant with “anchored by the partner manipulation” (as the partner manipulation was typically after the honesty oath manipulation or was that referring to Study 1?). We don’t think that this contributed to the null interaction effect as the order of the manipulations was not always the same across the different studies.

The order of different types of manipulations differs also across the study. For example, in Study 1 the honesty oath is presented before the minimal group paradigm. In Study 4 and 5 the first commitment to partner manipulation (group chat about group name) is before commitment to the oath. In the remaining studies, the oath is presented before the commitment to partner manipulations. Despite this variation, we are still able to find mostly consistent effects across studies.

We added a section in the limitations to address this issue:

“Fourth, due to the design of the studies we were unable to fully randomize the order of the different manipulations. In Studies 1-3 and 6-7 participants always committed to the honesty oath before being presented with the commitment to partner manipulation. In

Study 4 and 5 parts of the commitment to partner manipulation were presented before the honesty oath. A fully randomized design was difficult due to the fact that for Studies 2-7 the commitment to partner manipulation was part of the main task that also measured our dependent variable. This should be considered when interpreting the effect of both commitments.” (pp. 42-43)

I found the mind shape task to be quite clever and novel. However, as the authors indicated, the task did not allow the authors to capture the actual degree of cheating. As the authors move forward with this research, I encourage the authors to include at least one study that captures cheating in action.

We agree with the reviewer that it is a disadvantage that we are only able to measure dishonesty at the group- and not individual-level. However, there is good reason not to include a task that captures actual cheating as previous studies show that cheating rates are reduced considerably when participants are aware that their responses can be checked or tracked (Gerlach & Teodorescu, 2022; Schild et al., 2019). A possibility would have been to use deception, but we did not see the need to deceive participants given the possibility of using private group-level dishonesty indicators that are standard in the literature (Gerlach et al., 2019). However, we have added a short sentence to reflect on the reviewer’s good comment and our decision:

“We assessed dishonest behavior by using a variant of the so-called mind game (Study 1-5; e.g., Dimant et al., 2020; Jiang, 2013; Schild et al., 2019) or a die roll task (Study 2-3 & 6-7; Fischbacher & Föllmi-Heusi, 2013). In both tasks, participants have the opportunity to misreport their performance in order to maximize their payoffs. In both variants, the incentive for misreporting is high, as the actual choice occurs in private and cannot be verified by the researcher (Gerlach et al., 2019). These paradigms deliberately do not allow to measure dishonesty on the participant-level due to using a private task (but measure dishonesty on the group-level), as previous research has shown that employing public or traceable paradigms substantially reduces cheating rates (Gerlach & Teodorescu, 2022).” (pp. 12-13)

Minor notes:

p.4: Commitment to other individuals: within or outside of the organization? Might want to clarify

We have clarified this aspect that it refers to individuals within the organization.

Beneficiary matters? Dishonesty to benefit the self vs. benefit the group. See Emma Levine’s work for more on these distinctions.

We thank the reviewer for directing us at that literature and have added a reference to Levine’s work in the manuscript.

The authors try to replicate previous studies investigating the impact of social commitment (feeling committed to other persons) and commitment to social norms on dishonest behavior. They use two manipulations of social commitment (minimal group paradigm and live interaction) and an honesty oath to manipulate commitment to the social norm of telling the truth. Furthermore, they use two experimental tasks measuring the dependent variable lying behavior (mind game and die-in-the-cup task). The authors add to the previous literature the combination of social commitment and commitment to social norms. With more than 6,000 participants in pre-registered experiments, they find evidence that honesty oaths decrease dishonesty. They cannot replicate that commitment to other persons increases dishonesty. Finally, the authors observe a null effect when combining social commitment and commitment to social norms.

Given the replicability and credibility crisis in experimental psychology, more of these studies should be published. In my view, this applies also to the current study after the comments below are considered.

The limitations of the current study are no measurement of commitment in the baseline and oath treatments of Study 2, the manipulation checks of social commitment to other persons (in general), and the large variance of dishonesty across subject pools. However, the latter two limitations also apply to previous studies, and the limitations are mentioned.

Specific comments

1. In my view, the authors should elaborate a bit more on the limitations, especially the latter two. Furthermore, the pros and cons of using Prolific could be discussed.

We have elaborated more on both limitations and also provide a reflection of using Prolific as a participant pool.

“Although we employed a large variation of designs, tasks, payoffs, and populations there are still some limitations to note in the current studies. First, assessing the effectiveness of the commitment to other individuals was difficult, as it was not obvious which reference group should be used in the conditions in which no actual partner was involved. We also failed to assess felt commitment to the moral norm throughout the studies. We theorized that felt responsibility for one’s action might be a common mechanism of social commitment to other individuals and moral norms, with felt responsibility increasing with more commitment to the moral norm and decreasing with more commitment to other individuals. However, we failed to find any consistent effects. Future studies would need to assess commitment to both other individuals and moral norms more systematically to investigate possible mechanisms of the obtained effects. Relatedly, while our theoretical model assumes that participants felt social commitment to the experimenter when completing the oath, this was not explicitly mentioned. Future studies would need to make this aspect more salient, which should increase social commitment based on the theoretical assumption.” (p. 41)

“Third, dishonesty rates differed strongly across the different tasks and populations (see Gerlach & Teodorescu, 2022). It is essential to encourage dishonest behavior to be able to study it, which is why we employed private instead of public or traceable tasks. Some baseline dishonesty rates were rather low (e.g., $M = 3.72$ in Study 2), which might have influenced the effectiveness of our manipulations, particularly the honesty oath. Notwithstanding, an oath did significantly decrease dishonest behavior. This finding might also be related to the specific crowdsourcing populations we used. Participants, especially panelists in Study 2, might fear getting rejected for behaving dishonestly or being removed from the panel and not able to participate in future studies (Frollová et al., 2021). Studies would need to highlight the absence of punishment or study dishonesty in settings where fear of punishment is reduced as much as possible, while at the same time avoiding demand characteristics. The majority of studies were conducted on Prolific.co that has been considered to provide good data quality (Peer et al., 2017). On the other hand, these participants are also homogenous to a certain degree and have potentially taken part in numerous studies using similar paradigms or might act dishonestly because they think the experimenter wants them to (Houdek, 2017).” (p. 42)

2. I think the authors should more clearly disclose and discuss the relationship of the current study to the meta-analysis of Zickfeld et al. (2022).

We have added more information on how the current paper relates to the meta-analysis by Zickfeld et al. (2022). Originally, the meta-analysis was conducted before the current studies were performed but the meta-analysis was revised after the current paper was submitted and we included studies 1-6 in the meta-analysis.

“The current studies were conducted after a first version of the meta-analysis (Zickfeld et al., 2022) was submitted for publication. During the revision round of the meta-analysis all studies that had been conducted at that time (Study 1-6) were added to the meta-analysis based on an updated search. Study 7 was conducted after the meta-analysis was resubmitted for publication.” (Footnote 2)

3. Given their results, the authors should discuss in more detail the role of honesty oaths and how their design and findings contribute to previous literature. Some studies on honesty oaths are considered in Zickfeld et al. (2022) but not addressed in the current paper (e.g., Beck et al., 2020).

We have provided a more comprehensive review of the literature on honesty oaths and now include them in the Introduction and Discussion. We also elaborate on the discussion of how the current study contributes to previous studies by providing further evidence on the effectiveness of honesty oaths, though noting that the effect might be considerably small under certain conditions.

“As predicted (H2), we observed that committing to a moral norm by an honesty oath , reduced dishonest behavior to a degree that was small but not trivial, replicating previous studies (e.g., Beck, 2021; Beck et al., 2020; Dunaiev & Khadjavi, 2021; Heinicke et al., 2019; Jacquemet et al., 2018, 2020, 2021; Kanngiesser et al., 2021; Peer & Feldman, 2021). Effects were comparable across different countries and dishonesty measures, except for

Study 6. Although this finding was in the expected direction (logOR = -.23 [-.31, -.15], d = -.13), the effect size was considerably smaller than effects reported in previous meta-analyses (Bellé & Cantarelli, 2017; Zickfeld et al., 2022). We employed a widely used form of oath as in prior studies (e.g., Schild et al., 2019), varying only the timing of the oath in Study 1 and small parts of the formulation in Study 7, which did not have an effect on (dis)honesty. It is possible that different forms of expressing moral norm commitment (e.g., signing instead of copying text, though see Koning et al., 2020; Peer et al., 2023) and norms (e.g., descriptive rather than injunctive norms) might produce differential and possibly stronger effects (see Skowronek, 2022 for a framework to improve honesty nudges), but further studies are needed to confirm this effect. As previous studies did not find evidence for the effectiveness of commitment to oaths, especially in field settings (e.g., Kristal et al., 2020; Koning et al., 2020; Martuza et al., 2022), however, the present findings are reassuring. Nevertheless, it is important to study the conditions under which such an intervention could reduce dishonesty. For instance, we observed that the oath increased dishonesty in Study 6. Such reactance to oaths has been observed in previous studies (Cagala et al., 2019) and it would be interesting to investigate boundary conditions and the situational and personality variables that moderate such effects. A recent study suggested that oaths might work especially well for individuals that act honestly most of the time (Jacquemet et al., 2020), but we found the smallest association between trait honesty and reporting for participants committing to the oath. Recent studies have started to investigate boundary conditions of honesty oaths (e.g., Peer et al., 2023; Le Maux & Necker, 2023), but more standardized procedures and formulations are needed to make different studies comparable and evaluate why and when honesty oaths can be (in)effective or to what degree. Future studies would also need to investigate more applied settings in which individuals commit to an organizational moral norm (e.g., code of conduct) instead of a general honesty norm.” (pp. 38-39)

4. I think the graphs and tables need a better description and explanation in the text.

We have added a more detailed description for each table and figure in the text.

“Table 1 provides a specific overview of the power analysis and exclusion criteria for each study separately and Table 2 gives an overview of the different designs including the final sample size, the independent, and dependent variables. “ (p. 13)

“A distribution of the economic game scores for each study and treatment is provided in Figure 3 and the mean, standard deviation, cell sizes, and effect sizes for the different treatments are provided in Table 3. ” (p. 31)

“An overview of commitment scores to the partner for each treatment and study is provided in Figure 2.” (p. 29)

“A forest plot including the individual effects per experimental treatment and study is presented in Figure 4.” (p. 31)

5. I expected the Method section before the Results section.

We acknowledge that this was a misunderstanding, as the article was originally submitted to a different journal and automatically transferred to the current journal and we failed to update the specific section order.

In line with the guidelines of the current journal we have now moved the Method section before the Results section and think that this improves understanding considerably.

6. Although pre-registered, I was wondering about the author's approach to stop the recruiting once the main effects were statistically significant. I think this could be discussed in the paper.

We have added a short description why we adopted a sequential design in Studies 4-7. The advantages of a sequential design is that it is possible to conduct high-powered studies more efficiently and also to cancel studies when it becomes obvious that the expected effect was overestimated. More detailed information can for example be found in Lakens (2014). We don't think that we have space to discuss the pros and cons of conducting a sequential analysis in detail for the current manuscript.

"For Studies 4-7, we registered a sequential analysis approach in order to adopt a more efficient way to conduct a high-powered study given the small effects of the previous studies and to save resources (Lakens, 2014)." (p. 14)

7. The suggested numbers of independent observations (to achieve a power of 0.8) from the ex-ante power analyses differ substantially from the planned and finally realized numbers of independent observations – Study 1: 1060 vs. 770, Study 2: 2320 vs. 1494, Study 3: 7408 vs. 484, Study 4: 2160 vs. 1541, Study 5: 2160 vs. 982, Study 6: 1840 vs. 755. When justifying the sample size in the supplementary material, the authors seem to mix up the number of data points (observations) and the sample size (number of independent observations). What is the achieved power in the studies?

Yes, we were aware of the fact that our power analyses suggested substantially higher numbers than were finally registered as the planned sample size. This was based on several reasons. The first is that some of the studies were originally pilot studies for a different project. Second, as we used a repeated measures design for all studies (except Study 6), we expected the need for a smaller sample size than what was suggested. We did not find an easy solution to perform a power analysis for a multilevel logistic ordered regression, and wanted to reduce the likelihood of overpowering the present studies. We did not intend to mix up the number of data points and the number of independent observations, but it is most often the case that repeated measures approaches need fewer participants for the same power compared to one-shot designs (e.g., Guo et al., 2013).

As suggested by the reviewer and the editor, we have provided a sensitivity analysis to investigate the minimal detectable effect at 90% and 95% power for our six individual studies. We have added information on this in the main manuscript and detailed information is provided in the Supplement Section 2. Unfortunately, we did not find a solution on how to perform the sensitivity analysis for an ordinal logistic model and relied on a linear model for performing the sensitivity analyses.

“As final sample sizes across all studies were smaller than the suggested sample sizes by our power analyses, we conducted post-hoc sensitivity analyses for all studies in order to investigate what effect size we could minimally detect at a power of 90% and 95% (see Supplement Section 2.1). Across studies, the minimum effect size we could detect at 90% power ranged between $d = -.09$ and $-.19$ and between $d = -.10$ and $-.22$ for 95% power. The minimum effect size was larger in Study 6 (90% $d = -.30$; 95% $d = -.34$) due to the fact that we employed a one-shot game. These effect sizes are typically considered as small in the literature (Lovakov & Agadullina, 2021) and are in the range of our smallest effect size of interest ($d = +/- .15$). In addition, the meta-analytic investigation across 7,576 participants helps us to increase our power to detect even smaller effect sizes, such that we can be certain that we have enough power to detect our smallest effect size of interest when combining all studies.” (p. 15)

R3

The present work addresses how commitment to other individuals and to social norms influences dishonest behavior. In a set of six studies using different kinds of incentivized cheating paradigms, the authors systematically manipulated whether dishonest behavior affected another person’s outcome (social commitment: yes vs. no) and whether participants had to commit themselves to an oath (norm commitment: yes vs. no). Across studies, they found that commitment to others did not affect dishonesty, whereas commitment to an oath reduced dishonest behavior. In turn, commitment to an oath was less effective in decreasing dishonesty if participants were at the same time socially committed to another person.

Overall, I very much enjoyed reading this paper. It investigates an important and timely issue, it is well-written, and the methodology is sound. Moreover, I want to applaud the authors for their strong commitment to open science, having pre-registered all studies and providing all materials, data, and analysis scripts online. That said, however, I do have concerns about the framing of the paper and the general contribution. Specifically, I don’t think that the two factors the authors investigate should be considered instances of the same variable, that is, social commitment. Rather, the main contribution of the present paper seems to be that the authors are the first to combine two variables that have been shown to influence (dis)honest behavior. I detail these and a few additional concerns in what follows. In general, I don’t think that these issues can’t be healed in a revision, but it would probably require substantial rewriting.

1) The way how commitment is defined in the abstract (i.e., “feeling committed to other individuals or groups”) and on the very first page of the manuscript (i.e., “motivation to perform specific actions other agents are relying on”) does not fit the manipulation of social commitment via social norm compliance. Specifically, in the norm compliance condition, participants were asked to commit to an oath reading as follows: “Participants in this study commit to the norm of telling the truth. I promise that the information I am providing in this study is true”. Thus, in this condition, there is no other individual (or group) involved to which one can commit oneself – it is only about whether people commit to tell the truth, which may rather be considered a moral norm that is defined by the experimenters than a

generally held social norm. In this regard, I was also wondering about the “social commitment to other individuals” conditions. Here, dishonest behavior profits another participant by increasing their (monetary) outcome. Thus, dishonest behavior is prosocial, which is why one could likewise consider this manipulation to touch upon a social norm: the norm of prosociality. So, in general, I found the framing somewhat misleading and kept wondering about what exactly the authors manipulated here and what the studies can tell us. In my view, the major contribution of the present paper is that commitment to an oath is combined with manipulating the social justifiability of lying. This, however, would require a different framing of the manuscript and a major overhaul of the reasoning in general.

We have revised the introduction and our theoretical framework to make clear why we think that both commitment to another individual and commitment to a moral norm are instances of the same variable (social commitment).

First, we have changed the formulation of commitment to *social norm* to commitment to *moral norm*, which is more in line with previous research (e.g., Hertwig & Mazar, 2022) and also addresses the point rightfully raised by you that wanting to cooperate with another person can be considered a social norm as well. We think this makes our focus clearer. At the same time, we still consider this moral norm to be a social norm as well and part of the same basic mechanism of social commitment.

Even though participants commit to tell the truth (to a moral norm, which we disagree with that it is defined by us as the experimenters, but rather an injunctive norm), we argue that committing to a moral norm always takes place in a social and cultural environment. That is, the oath taker commits themselves not only to telling the truth, but does so by committing to other people that are relying on the person to tell the truth (e.g., the experimenter, an organization, society).

We now provide a detailed definition of what we consider social commitment and provide examples why we consider both commitment to a moral norm and commitment to another individual to follow the same underlying process.

“Social commitment has been defined in various ways, focusing on specific commitments to other individuals (Arriaga & Agnew, 2001) or commitment to organizations (Cho & Park, 2011; Reichers, 1985). Here, we adopt a minimal definition of social commitment arguing that social commitment refers to the dispositional state of an agent (X), who is motivated to carry out an action (z) because some other agent(s) (Y) is relying on them to do so (Michael et al., 2016; Zickfeld et al., 2022). Thus, social commitment can refer to a specific action or a general motivational dispositional state of agent X in relationship to agent Y. It is also sufficient that X thinks that Y is relying on the specific action. In contrast to self-commitment (e.g., committing to the goal of eating less candy), social commitment always involves at least two agents. Social commitment can further be described as unilateral, only one agent feeling committed to the other, and mutual, both agents feeling committed to each other by the same (joint) or different (complementary) goal (Clark, 2020). ” (p. 5)

Further, we try to explain why we consider commitment to a moral norm based on the same basic mechanism as commitment to another individual.

“In the case of social commitment to moral norms, a typical task is asking participants to commit to an honesty oath (e.g., Beck et al., 2021; Jacquemet et al., 2020; Hertwig & Mazar, 2022). In this case, the participant (X) takes the honesty oath by for example signing, checking, or copying it (see Peer et al., 2023) and is motivated to accurately report their outcome in a task (z) since the party representing the oath (Y), in the context of experimental studies commonly the experimenter, relies on this action. This creates a unilateral commitment to the experimenter on the part of the participant. The experimenter relies on the participant accurately reporting their outcomes because dishonest behavior on the part of the participant will result in financial losses (and violate a social norm of honesty). Manipulating the commitment to a moral norm (e.g., via an honesty oath) should increase the motivational state to perform an action (i.e., reporting accurately) because another agent relies on it, that is social commitment (Zickfeld et al., 2022). Of course, it can be argued that committing to an honesty oath also represents a form of self-commitment in which individuals commit to their future self in telling the truth. However, we emphasize that honesty oaths represent social institutions (Mercier & Boyer, 2021; Rutgers, 2013) and that nonobservance has direct negative consequences on other individuals, organizations, or society at large (de Bruin, 2016). Therefore, commitment to an honesty oath can be considered a social norm, as other individuals in a given community or society expect and demand of each other to follow it and do so with a certain prevalence (see Malle, 2023). This emphasizes the view that social commitment to moral or social norms is embedded in every social, cultural, and interpersonal environment (Zickfeld et al., 2022). We will now turn to the question how social commitment can influence dishonest behavior.” (pp. 6-7)

We assessed social commitment only in response to the partner, but not in response to the agent representing the oath. We have added this as a limitation to the discussion.

“Future studies would need to assess commitment to both other individuals and moral norms more systematically to investigate possible mechanisms of the obtained effects. Relatedly, while our theoretical model assumes that participants felt social commitment to the experimenter when completing the oath, this was not explicitly mentioned. Future studies would need to make this aspect more salient, which should increase social commitment based on the theoretical assumption. “ (p. 41)

2) I think the surprisingly weak effect observed for commitment to another individual may be attributable to the specific manipulation of social commitment used. Specifically, participants knew that their own and the other's individual outcomes would be summed up and then distributed among the two. Thus, both acted independently of each other and both their behavior influenced each person's outcome. Arguably, previous research showing an effect of commitment to others on dishonesty used stronger manipulations, for example, by having participants work together to increase their joint outcome (i.e., collaborative corruption) or by having another party profit from the individual's dishonesty while this other party had no opportunity to obtain a reward themselves (i.e., they did not participate in the cheating task). In the present studies, the felt obligation to help the other (i.e., the

partner) was thus likely lower than in previous studies using different manipulations. Is there any evidence that the authors can refer to to test this idea or show the comparability of their manipulation of social commitment with other manipulations? This would greatly help assess the conclusiveness of the findings at hand.

We agree with the reviewer that our commitment to another individual manipulations can be considered as relatively weak with regard to real-life interactions or repeated exposures. Nevertheless, it was based on previous studies that have found (much stronger) effects on dishonesty (e.g., Conrads et al., 2013; Wiltermuth, 2011). We respectfully disagree that “both acted independently of each other” as indeed the final outcome in Studies 2-6 depended on the report of the partner. So, while each participant reported independently, their final outcome was not independent.

Based on the reviewer’s suggestions, we have added another study (Study 7) in which we employ the original sequential die roll paradigm by Weisel and Shalvi (2015) that is likely inducing stronger commitment. Nevertheless, we obtain similar effects to the previous studies finding only a small effect of commitment to other individuals on dishonesty that is statistically not significant. This speaks to the robustness of our overall findings that we find no credible evidence that commitment to other individuals increases dishonesty using typical online paradigms.

3) Hypothesis 3 suggests that the effects of the two manipulated factors are similar in size. Is this assumption really justified based on previous evidence? I think more needs to be done to justify that combining the two manipulations should indeed yield a zero effect because they will cancel each other out.

We have now improved our theoretical and empirical argument why we predict a null effect for H3. We argue that commitment to moral norms and individuals at the same time activates concerns to both honesty and prosocial norms, which cancel each other out. Based on findings from a recent meta-analysis (Zickfeld et al., 2022), findings were indeed comparable in opposite direction and we predicted that these will cancel each other out. We now write:

“From a theoretical perspective, we would expect that social commitment to other individuals and moral norms at the same time, induces attendance to both prosocial norms on the one hand (i.e., wanting to help the other individual or the group) and attendance to moral norms on the other hand (i.e., wanting to be an honest person). Whether one of these can override the other might depend on the salience of each concern or norm (i.e., the strength of the (social) commitment). In the Zickfeld and colleagues (2022) meta-analysis both effects were similar in strength ($g = -.17$ vs. $g = .24$), suggesting that combining both commitments might cancel each other out. Therefore, we predict that:

H3. Combining social commitment to individuals and to an honesty oath (compared to the baseline) does neither increase nor decrease dishonest behavior (null effect).” (p. 11)

More minor comments:

4) It would be useful to provide at least a brief summary of the methods at the end of the

introduction. Otherwise, it is very difficult to follow the remainder of the paper without reading the Methods section first.

Based on the journal guidelines we have rearranged the order of sections and now the Method section is first facilitating a comprehension of the results.

5) It should be specified that Figure 2 refers to commitment to the partner, rather than commitment to the oath.

We have added this information.

6) The abbreviation "IOS" (in the Design & Procedure section) is not defined.

We have provided information that this refers to the inclusion-of-the-other-in-the-self measure.

7th Jul 23

Dear Dr Zickfeld,

Your manuscript titled "Investigating the Impact of Social Commitment on Dishonest Behavior" has now been seen by Reviewers #2 and #3, whose comments appear below. Reviewer #1 was unable to return a report.

I am delighted to say that we are happy, in principle, to publish a suitably revised version of your work in Communications Psychology under the open access CC BY license (Creative Commons Attribution v4.0 International License).

We invite you to revise your paper one last time to address the remaining editorial requests. At the same time we ask that you edit your manuscript to comply with our format requirements and to maximise the accessibility and therefore the impact of your work.

EDITORIAL REQUESTS:

SUBMISSION INFORMATION:

OPEN ACCESS:

Communications Psychology is a fully open access journal. Articles are made freely accessible on publication under a [CC BY license](http://creativecommons.org/licenses/by/4.0) (Creative Commons Attribution 4.0 International License). This license allows maximum dissemination and re-use of open access materials and is preferred by many research funding bodies.

For further information about article processing charges, open access funding, and advice and support from Nature Research, please visit <https://www.nature.com/commspsychol/article-processing-charges>

At acceptance, you will be provided with instructions for completing this CC BY license on behalf of all authors. This grants us the necessary permissions to publish your paper. Additionally, you will be asked to declare that all required third party permissions have been obtained, and to provide billing

information in order to pay the article-processing charge (APC).

* **DATA AVAILABILITY:**

[link redacted]

Best regards,

Marike

Marike Schiffer, PhD
Chief Editor
Communications Psychology

REVIEWERS' COMMENTS:

Reviewer #2 (Remarks to the Author):

The authors did a very good job in responding to all of my previous comments. Thank you!

Reviewer #3 (Remarks to the Author):

I have been a reviewer of this paper before (R3) and I want to thank the authors for seriously considering my comments and addressing them accordingly. Overall, I think the authors did a great job with the revision, even collecting additional data, and the paper has considerably improved as a consequence. Thus, it is my pleasure to recommend this paper for publication in its current form.